# Contact Map Transfer with Conditional Diffusion Model for Generalizable Dexterous Grasp Generation

**Yiyao Ma,    Kai Chen[†],    Kexin Zheng,    Qi Dou**

Department of Computer Science and Engineering,
The Chinese University of Hong Kong

## Abstract

Dexterous grasp generation is a fundamental challenge in robotics, requiring both grasp stability and adaptability across diverse objects and tasks. Analytical methods ensure stable grasps but are inefficient and lack task adaptability, while generative approaches improve efficiency and task integration but generalize poorly to unseen objects and tasks due to data limitations. In this paper, we propose a transfer-based framework for dexterous grasp generation, leveraging a conditional diffusion model to transfer high-quality grasps from shape templates to novel objects within the same category. Specifically, we reformulate the grasp transfer problem as the generation of an object contact map, incorporating object shape similarity and task specifications into the diffusion process. To handle complex shape variations, we introduce a dual mapping mechanism, capturing intricate geometric relationship between shape templates and novel objects. Beyond the contact map, we derive two additional object-centric maps, the part map and direction map, to encode finer contact details for more stable grasps. We then develop a cascaded conditional diffusion model framework to jointly transfer these three maps, ensuring their intra-consistency. Finally, we introduce a robust grasp recovery mechanism, identifying reliable contact points and optimizing grasp configurations efficiently. Extensive experiments demonstrate the superiority of our proposed method. Our approach effectively balances grasp quality, generation efficiency, and generalization performance across various tasks. Project homepage: https://cmtdiffusion.github.io/

## 1 Introduction

Dexterous grasp generation is a crucial task in robotics, enabling robotic hands to manipulate objects with human-like precision and adaptability [1, 2, 3]. It plays a fundamental role in humanoid robotics, particularly in its application in unstructured environments[4, 5] and human-robot interactions [6, 7, 8, 9]. However, the high degrees of freedom of dexterous robotic hands increase the difficulty of achieving stable grasps [1], while complex contact dynamics between robot fingers and objects in diverse shapes hinder the generalization of existing methods [10, 11]. How to generate stable and generalizable grasps across various objects remains a challenging problem that has gained extensive attention [12, 13, 14, 15] in recent years but has not yet been fully resolved.

Existing dexterous grasp generation methods can be broadly categorized into analytical [16, 17, 18, 19, 20, 21] and generative [22, 23, 24, 25, 26, 27] approaches. Analytical methods optimize grasps by defining objective functions based on hand-object constraints such as force closure [16, 17, 20], penetration [21], and hand coverage [18, 19], ensuring high-quality and stable grasp synthesis.

---

[†]Corresponding author
{yyma23, kaichen}@cse.cuhk.edu.hk

39th Conference on Neural Information Processing Systems (NeurIPS 2025).

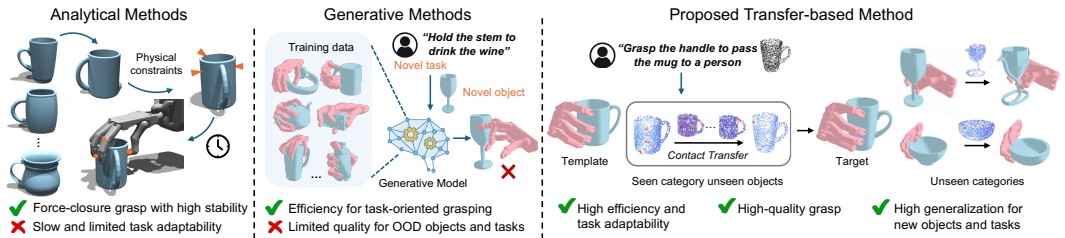

Figure 1: Comparison of our proposed framework with existing analytical and generative methods. The proposed transfer-based framework can effectively balance efficiency, quality, task adaptability, and generalization capability for dexterous grasp generation.

However, these methods are computationally expensive due to the complexity of the optimization process [28]. Moreover, generating task-oriented grasps requires manually designing task-specific constraints (*e.g.*, object part constraints [29] or wrench constraints [30]), making them difficult to generalize across diverse real-world robotic tasks. In contrast, generative methods learn grasp distributions conditioned on object features, enabling efficient and diverse dexterous grasp synthesis. By integrating task embeddings [31, 32, 33], they can further produce grasps aligned with specific task requirements. However, their performance heavily depends on the quality of training data, including the diversity of objects, accuracy of grasp annotations, and completeness of task descriptions. As a result, generative models still struggle to generalize to out-of-distribution (OOD) objects and novel tasks, where the learned grasp distribution may fail to transfer effectively. As illustrated in Fig. 1, while analytical methods ensure grasp stability and generative methods enhance efficiency and task adaptability, no existing approach effectively integrates their strengths to achieve both stability and generalization in dexterous grasp generation.

In this regard, we innovate a transfer-based framework for dexterous grasp generation. The core idea is to use generative models to efficiently transfer high-quality grasps sampled by analytical methods on a shape template to novel objects within the same category. Unlike conventional analytical methods, our approach can improve efficiency and adaptability by using generative models for grasp transfer. Compared to existing generative approaches, our method models the conditional grasp distribution based on task embeddings and geometric similarities across object categories, rather than directly learning an implicit distribution tied to object features [31, 32, 33]. This enables the generation of high-quality, task-oriented grasps while enhancing generalization to new object instances, unseen tasks, and even novel object categories. Nevertheless, implementing this framework presents several challenges. The substantial shape variations across objects, the complex contact interactions between robot hands and diverse object geometries, and the need to accommodate varying task specifications introduce fundamental difficulties that must be addressed.

In this paper, we present a novel conditional diffusion model for dexterous grasp transfer. Specifically, we follow [34, 22, 35, 36] and leverage robust object-centric representations to reformulate the grasp transfer problem as the generation of an object contact map. We then propose a framework that integrates object geometry similarity along with the task embedding into the diffusion model for a conditional generation of the contact map. To handle the complex shape variations across diverse objects, we introduce a dual mapping mechanism within conditional diffusion. This mechanism explicitly models the dual mapping relationships between the template shape and the reconstructed template contact map, and between the noise and the contact map of the novel object. In this way, the diffusion model effectively captures the intricate geometric similarities between the shape template and the novel object under different task specifications, enabling the generation of an accurate contact map aligned with the intended grasping task.

The contact map alone indicates whether the dexterous robotic hand is in contact with the object but fails to capture the finer details of the contact interaction. To achieve more stable and dexterous grasping, we follow [35] to further derive two additional object-centric maps, the part map and the direction map, which provide richer information about the contact regions and the grasping orientations. To jointly transfer three object-centric maps from the shape template to the novel object, we develop a cascaded conditional diffusion framework. This cascaded design enables a progressive generation process, ensuring intra-consistency among the three maps and preserving coherent contact, part, and direction information throughout the transfer process. Based on the transferred three object-

centric maps, we design a robust mechanism to automatically identify object points with reliable part and direction predictions. Finally, we recover the grasp configuration parameters through a fast and robust optimization scheme, ensuring the stability and feasibility of the generated dexterous grasps for novel objects. We summarize our main contributions as follows:

- We formulate dexterous grasp generation as an object contact map transfer problem and propose a novel conditional diffusion model that jointly captures object geometric similarity and textual task embeddings, enabling more generalizable dexterous grasp generation.
- We exploit object-centric part map and direction map to enrich hand-object contact representation and develop a cascaded conditional diffusion framework to jointly transfer the object contact map, part map, and direction map with high consistency.
- We propose a robust optimization method that adaptively identifies object points with reliable part and direction predictions and comprehensively leverages contact information from the contact, part, and direction maps to robustly recover dexterous grasp parameters.
- Extensive experiments show that our method can transfer dexterous grasps from shape templates to novel objects with high quality. It generalizes well across diverse objects, tasks, and unseen categories while generating grasps that well align with task specifications.

## 2   Related Work

**Analytical Methods for Dexterous Grasp Generation.** This kind of methods typically formulates dexterous grasp generation as an optimization problem, leveraging analytical models of contact mechanics [18, 19], force closure [37, 16, 17], and grasp stability [21, 38] to compute optimal hand configurations for grasping a given object. While previous methods [39, 40, 41, 37] rely on computationally expensive approaches to compute force closure between the robotic hand and the object for grasp optimization, recent methods [16, 17, 20] have introduced differentiable force-closure estimators to speed up this computation. In addition, to apply these analytical methods to task-oriented grasping, recent approaches define task-specific object parts [29] for grasping or constrain the wrench relationship [30] between the robotic hand and the object based on the task type, thereby generating dexterous grasps that align with task requirements. However, these heuristic methods still struggle to generalize across diverse objects and tasks. In this paper, we will explore a more efficient and generalizable transfer-based framework for dexterous grasp generation.

**Generative Methods for Dexterous Grasp Generation.** Generative methods for dexterous grasp generation [42, 43] learn grasp distributions conditioned on object features, typically through direct parameter prediction [24, 44, 45] or optimization using object-centric contact representations [46, 22, 23, 35]. Compared to analytical methods, these methods can effectively incorporate high-level task embeddings [47, 48, 49, 32, 31, 50, 25], making them well-suited for practical applications. Among such methods, diffusion models [51, 52] are particularly adept at modeling the complex data distributions [53, 54, 55, 56, 57], and recent works [58, 59, 60, 61, 62, 63, 33] have explored diffusion-based methods to improve quality and efficiency for dexterous grasp generation. For example, DexDiffuser [58] generates dexterous grasps by denoising randomly sampled grasp pose parameters conditioned on object point clouds. FastGrasp [59] introduces a one-stage diffusion model to enhance grasp generation efficiency and incorporates physical constraints into the latent grasp representation to improve grasp quality. UGG [62] further develops a unified diffusion model for multi-task learning of hand-object interactions. However, the performance of generative methods is often limited by training data quality and scale, potentially yielding less stable grasps than analytical methods. This paper proposes a diffusion-based grasp transfer framework that leverages object geometric similarity via text embeddings, instead of directly learning grasp distributions from training data. Our framework offers high scalability and effectively utilizes high-quality grasps from shape templates to enhance grasp quality for novel objects.

**Transfer-based Methods for Dexterous Grasp Generation.** Previous transfer-based methods either necessitate training on individual object category [27, 64], or depend on pre-defined object parts and functionalities to guide the transfer process [26, 65], which limit their ability to generalize effectively to novel tasks and categories. Recent methods employ detailed object shape representations, such as Neural Radiance Fields (NeRF) or Signed Distance Functions (SDFs), to enable higher-quality grasp transfer [66, 67, 68, 69, 70]. For instance, Tink [69] learns an implicit SDF for each object instance and transfers grasps via shape interpolation. However, it requires learning separate models for each

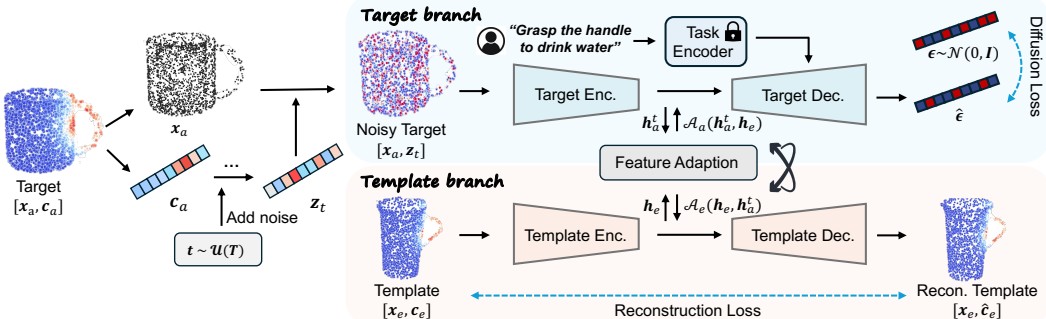

Figure 2: Our conditional diffusion model learns to transfer contact maps via a template-target framework. The template branch encodes shape templates $(\mathbf{x}_e, \mathbf{c}_e) \rightarrow \mathbf{h}_e$, while the target branch denoises latent vector $\mathbf{z}_t$ conditioned on target geometry $\mathbf{x}_a$, template features $\mathbf{h}_e$, and language $\ell$. A bidirectional adaptation module $\mathcal{A}$ bridges these branches through feature integration.

object or category, which is computationally expensive and fundamentally limits generalization to novel categories. In this paper, we introduce a conditional diffusion model for grasp transfer, which jointly learns task specifications and the shape similarity between a given shape template and novel objects. After training on a set of base categories, our approach exhibits strong generalization to novel categories and tasks without the need for per-category retraining.

## 3   Methodology

In this section, we introduce our approach for dexterous grasp generation. Given a **target** object $\mathbf{x}_a \in \mathbb{R}^{m \times 3}$ represented by $m$ points, our goal is to generate the corresponding dexterous grasp configuration $\mathbf{q_a} \in \mathbb{R}^k$ based on the task specification $\ell$, where $k$ denotes the degree-of-freedom of the robot hand (*e.g.*, $k = 24$ for the ShadowHand). To ensure robust grasp recovery, we follow the pipeline [22, 35], which first generates an object contact map $\mathbf{c} \in \mathbb{R}^{m \times 1}$ and then optimizes the grasp parameters based on the contact information. We formulate this process as a contact transfer problem. Specifically, let $\mathbf{x}_e \in \mathbb{R}^{n \times 3}$ be a shape **template** associated with $\mathbf{x}_a$, and let $\mathbf{c}_e$ denote its contact map for task $\ell$. Our objective is to transfer $\mathbf{c}_e$ from $\mathbf{x}_e$ to $\mathbf{x}_a$ to obtain the target contact map $\mathbf{c}_a$ that aligns with $\ell$, which then would be used to recover the grasp configuration.

To this end, we first introduce a conditional diffusion model in Sec. 3.1, which jointly integrates the geometric similarity between $\mathbf{x}_a$ and $\mathbf{x}_e$ with the textual task description for contact map transfer. In Sec. 3.2, we further present an object-centric part and direction mapping mechanism to enhance contact representation and present a cascaded framework based on the conditional diffusion model to jointly transfer object contact, part, and direction maps while ensuring their mutual consistency. Finally, in Sec. 3.3, we develop a robust optimization method that adaptively identifies reliable object points based on part and direction predictions, efficiently leveraging the enriched contact information to recover dexterous grasp parameters with high robustness.

### 3.1   Conditional Diffusion Model for Contact Map Transfer

**Diffusion Preliminaries.** A denoising diffusion probabilistic model (DDPM) is trained to learn the transformation relationship between $\mathbf{c}_a$ and a latent variable $\mathbf{z} \in \mathbb{R}^{m \times 1}$ sampled from a tractable Gaussian distribution. Formally, this learning process consists of two Markov chains: a forward chain $q(\mathbf{z}_t | \mathbf{c}_a) = \mathcal{N}(\mathbf{z}_t; \sqrt{\overline{\alpha}_t} \mathbf{c}_a, (1 - \overline{\alpha}_t)\mathbf{I})$ that gradually adds noise to $\mathbf{c}_a$ over $T$ timesteps, where $\overline{\alpha}_t := \prod_{s=1}^{t} \alpha_s$ is the cumulative product of noise scheduling coefficients $\alpha_s \in (0, 1)$, and $\mathbf{z}_t$ represents the noisy version of $\mathbf{c}_a$ at timestep $t$. The reverse process $p_\theta(\mathbf{c}_a | \mathbf{z}_t, C)$ iteratively reconstructs $\mathbf{c}_a$ from $\mathbf{z}_t$ through a noise prediction network $\boldsymbol{\epsilon}_\theta(\mathbf{z}_t, C, t)$, where $C$ is the condition on $\mathbf{x}_e, \mathbf{x}_a, \mathbf{c}_e, \ell$. The model is trained by minimizing the objective:

$$\mathcal{L}(\theta) = \mathbb{E}_{t \sim \mathcal{U}(1,T), \boldsymbol{\epsilon} \sim \mathcal{N}(0,\mathbf{I})} \| \boldsymbol{\epsilon} - \boldsymbol{\epsilon}_\theta(\mathbf{z}_t, C, t) \|^2, \tag{1}$$

where $\mathcal{U}$ and $\mathcal{N}$ are the uniform and Gaussian distributions, respectively. While this vanilla conditional diffusion implicitly captures the distribution of contact maps, it would fall short in handling the

shape variation between $\mathbf{x}_e$ and $\mathbf{x}_a$ for contact map transfer. We therefore present a conditional diffusion model with a dual mapping architecture to enhance the geometric feature learning of the shape template and novel target objects, and to improve the contact map transfer performance.

**Diffusion with Dual Mapping Branch.** As shown in Fig. 2, we propose a diffusion architecture consisting of a template branch and a target branch to jointly learn template feature $\mathbf{h}_e$ and target feature $\mathbf{h}_a^t$, coupled with a dual feature adaptation module $\mathcal{A}$ that mutually integrates the learned features to bridge the shape gap between them. For the template branch, we employ a reconstruction network to encode and reconstruct the template's contact map. First, the point encoder $f_{\text{enc}}$ extracts geometric and semantic features $\mathbf{h}_e = f_{\text{enc}}(\mathbf{x}_e, \mathbf{c}_e)$ from the template's point cloud and its ground-truth contact map. These features are then adapted by the adaptation module $\mathcal{A}_e$, which injects target-aware characteristics from the target branch's intermediate features $\mathbf{h}_a^t$. Finally, the network is trained to minimize the reconstruction loss:

$$\mathcal{L}_{\text{recon}} = ||\mathbf{c}_e - f_{\text{dec}}(\mathcal{A}_e(\mathbf{h}_e, \mathbf{h}_a^t))||^2. \tag{2}$$

For the target branch, we implement the denoising network $\boldsymbol{\epsilon}_\theta$ comprising the encoder $g_{\text{enc}}$ and decoder $g_{\text{dec}}$, where $\boldsymbol{\epsilon}_\theta(\mathbf{z}_t, C, t) \triangleq g_{\text{dec}}\big(\mathcal{A}_a(g_{\text{enc}}(\mathbf{x}_a, \mathbf{z}_t), \mathbf{h}_e), f_l(\ell), t\big)$ explicitly combines geometric encoding and conditional noise prediction. In specific, we first extract target features $\mathbf{h}_a^t = g_{\text{enc}}(\mathbf{x}_a, \mathbf{z}_t)$, fuse them with template features $\mathbf{h}_e$ via adaptation module $\mathcal{A}_a$, and condition on language features $f_l(\ell)$ [71]. The target decoder $g_{\text{dec}}$ ultimately predicts the noise to optimize the following objective:

$$\mathcal{L}_{\text{diff}} = \mathbb{E}_{t \sim \mathcal{U}(1,T), \boldsymbol{\epsilon} \sim \mathcal{N}(0,\mathbf{I})} ||\boldsymbol{\epsilon} - \boldsymbol{\epsilon}_\theta(\mathbf{z}_t, C, t)||^2. \tag{3}$$

**Adaptation Module for Conditional Diffusion.** The adaptation module $\mathcal{A}$ is designed to mutually integrate features between the template and target branches, serving as a conditioning mechanism to guide the learning process for both networks. Formally, the adaptation process for $\mathcal{A}_a$ is defined as:

$$\mathcal{A}_a(\mathbf{h}_a^t, \mathbf{h}_e) = \text{MLP}\left(\text{softmax}\left(\frac{\mathbf{h}_a^t \mathbf{h}_e^T}{\sqrt{d}}\right)\mathbf{h}_e\right) + \mathbf{h}_a^t, \tag{4}$$

where $d$ is the dimension of $\mathbf{h}_e$. Similarly, $\mathcal{A}_e$ operates in reverse direction, merging the target feature to template. By adaptively attending to the relevant regions of the template (target) feature based on the target (template) geometric and semantic context, the model can effectively learn the inherent relationship between shape templates and novel objects.

**Training and Sampling.** During training, the model jointly optimizes the parameters of the template and target branches. The overall training objective combines the reconstruction loss from template branch and the diffusion loss from the target branch $\mathcal{L}_{\text{contact}} = \mathcal{L}_{\text{recon}} + \lambda \mathcal{L}_{\text{diff}}$, where $\lambda = 1$ is a weighting factor. During inference, we first sample pure noise $\mathbf{z}_T \sim \mathcal{N}(0, \mathbf{I})$ and iteratively denoise it through $T$ steps to generate the target contact map $\hat{\mathbf{c}}_a \triangleq \mathbf{z}_0$. At each timestep $t = T, \ldots, 1$, the latent state updates as $\mathbf{z}_{t-1} = \mu_\theta(\mathbf{z}_t, C, t) + \sigma_t \boldsymbol{\epsilon}$ with $\boldsymbol{\epsilon} \sim \mathcal{N}(0, \mathbf{I})$, here $\sigma_t$ determines the stochastic noise injection magnitude during denoising, following the variance schedule as derived in DDPM. The conditional mean $\mu_\theta$ is computed by the target denoising network $\boldsymbol{\epsilon}_\theta$ through:

$$\mu_\theta(\mathbf{z}_t, C, t) = \frac{1}{\sqrt{\alpha_t}}\left(\mathbf{z}_t - \frac{1 - \alpha_t}{\sqrt{1 - \bar{\alpha}_t}}\boldsymbol{\epsilon}_\theta(\mathbf{z}_t, C, t)\right), \tag{5}$$

with $C \triangleq \{\mathcal{A}_a(g_{\text{enc}}(\mathbf{x}_a, \mathbf{z}_t), f_{\text{enc}}(\mathbf{x}_e, \mathbf{c}_e)), f_l(\ell)\}$ combining all conditions for generation.

## 3.2 Cascaded Conditional Diffusion for Joint Contact, Part, and Direction Transfer

**Part and Direction Map for Dexterous Grasp.** Despite the contact map provides a useful representation of contact regions, it alone is insufficient to fully capture the complex hand-object interaction, leaving ambiguities in both the specific grasping parts and the grasping style. To address this, we follow a previous work [35] on human grasp synthesis and model dexterous grasps using additional part maps and direction maps. The part map $\mathbf{p} \in \mathbb{R}^{n \times b}$ indicates the closest hand part for each point on the object, where $b = 16$ represents the 16 predefined parts of the dexterous hand. The direction map $\mathbf{d} \in \mathbb{R}^{n \times 3}$ encodes the direction from each point on the object to the center of its corresponding hand part. Together, these object-centric contact, part, and direction maps provide a more precise and detailed representation of the dexterous grasp, enhancing the potential for robust grasp transfer.

**Cascaded Diffusion Framework.** Since the three maps collectively describe different aspects of the same hand-object interaction, it is essential to ensure the inherent consistency during transfer. In this regard, we develop a cascaded diffusion framework as shown in Fig. 3 to introduce the previously generated maps as additional conditions to guide the generation of subsequent maps.

Specifically, the generation of the target's part map $\mathbf{p}_a$ is conditioned on $(\mathbf{x}_e, \mathbf{c}_e, \mathbf{p}_e, \mathbf{x}_a, \hat{\mathbf{c}}_a)$, where $\mathbf{p}_e$ is the template's part map and $\hat{\mathbf{c}}_a$ is the predicted target's contact map. Similar to contact diffusion, we jointly optimize both re-

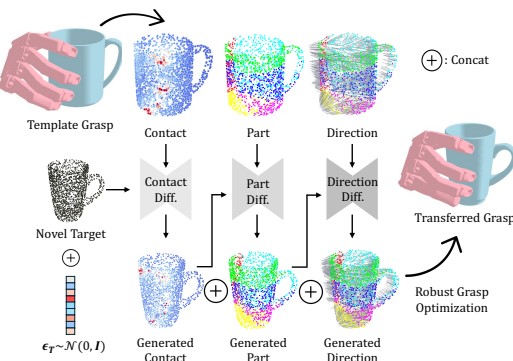

Figure 3: An overview of the cascaded diffusion framework.

construction and diffusion networks. However, since the part map is represented by $b$ discrete labels, the template branch predicts the probability distribution over the $b$ parts for each point, and the training is supervised using a negative log-likelihood (NLL) loss:

$$\mathcal{L}_{recon}^{part} = -\frac{1}{n} \sum_{i=1}^{n} \sum_{j=1}^{b} \mathbf{p}_e^{(i,j)} \log \hat{\mathbf{p}}_e^{(i,j)}, \tag{6}$$

where $\mathbf{p}_e^{(i,j)}$ and $\hat{\mathbf{p}}_e^{(i,j)}$ are the ground-truth and predicted association probabilities between point $i$ and part $j$. For the direction map, we employ cosine similarity to measure the alignment between the reconstructed direction map $\hat{\mathbf{d}}_e$ and the ground truth $\mathbf{d}_e$:

$$\mathcal{L}_{recon}^{dir} = -\frac{1}{n} \sum_{i=1}^{n} \frac{\mathbf{d}_e^{(i)} \cdot \hat{\mathbf{d}}_e^{(i)}}{\|\mathbf{d}_e^{(i)}\|\|\hat{\mathbf{d}}_e^{(i)}\|}, \tag{7}$$

where $\mathbf{d}_e^{(i)}$ and $\hat{\mathbf{d}}_e^{(i)}$ are the ground-truth and predicted direction vectors for point $i$. Please refer to Sec. A.1 for more details on the cascaded diffusion models.

## 3.3 Robust Dexterous Grasp Optimization with Transferred Maps

**Identify Object Points with Reliable Map Values.** Given the transferred contact map $\mathbf{c}$, part map $\mathbf{p}$, and direction map $\mathbf{d}$ for a novel object, we develop a method to identify object points with reliable map values to improve the robustness and accuracy of the recovered grasp parameters. By definition, the part map $\mathbf{p}$ assigns each object point to a hand part, while the direction map $\mathbf{d}$ provides a unit vector indicating the intended contact direction for grasping. Notably, for all points belonging to the same part in $\mathbf{p}$, their corresponding direction vectors in $\mathbf{d}$ should theoretically converge at a single point in space. This point represents the closest joint position on the robotic hand associated with the grasping action for that object part.

Leveraging this property, we can estimate the corresponding joint position for each part in $\mathbf{p}$ by computing the intersection of direction vectors within the same part. Specifically, let $\mathbf{x} = \{\mathbf{x}^i | i = 1, ..., w\} \in \mathbb{R}^{3 \times w}$ be a part with $w$ points, and $\mathbf{d}_\mathbf{x} = \{\mathbf{d}_\mathbf{x}^i | i = 1, ..., w\} \in \mathbb{R}^{3 \times w}$ denote the associated normalized directions. The corresponding joint position $J$ can be recovered by minimizing:

$$D(J; \mathbf{x}, \mathbf{d}_\mathbf{x}) = \sum_{i=1}^{w} (\mathbf{x}^i - J)^\top (\mathbf{I} - \mathbf{d}_\mathbf{x}^i (\mathbf{d}_\mathbf{x}^i)^\top)(\mathbf{x}^i - J), \tag{8}$$

which computes the sum of squared distance between $J$ and the normalized direction vectors. Given that $\partial D / \partial J = \sum_{i=1}^{w} -2 \times (\mathbf{I} - \mathbf{d}_\mathbf{x}^i (\mathbf{d}_\mathbf{x}^i)^\top)(\mathbf{x}^i - J)$, optimizing Eq. 8 equals to solve a linear function as $\mathbf{A} \times J = \mathbf{b}$, where $\mathbf{A} = \sum_{i=1}^{w} (\mathbf{I} - \mathbf{d}_\mathbf{x}^i (\mathbf{d}_\mathbf{x}^i)^\top)$, and $\mathbf{b} = \sum_{i=1}^{w} (\mathbf{I} - \mathbf{d}_\mathbf{x}^i (\mathbf{d}_\mathbf{x}^i)^\top) \mathbf{x}^i$. Then, the joint position can be solved by $\hat{J} = \mathbf{A}^\dagger \mathbf{b}$, where $\mathbf{A}^\dagger$ denotes the Moore-Penose pseudoinverse of $\mathbf{A}$. Based on the recovered joint positions, we apply two filtering rules to select reliable object points for grasp parameter optimization: (i) The average distance between $\hat{J}$ and $\mathbf{x}$ should smaller than a threshold $\tau_a$. This criterion eliminates parts where the direction predictions are highly noisy. (ii) Each object point in $\mathbf{x}$ should lie within a distance threshold $\tau_b$ from the joint position $J$. This rule removes outliers in the part map where individual points significantly deviate from the expected grasping region.

| Methods | Analytical | | Generative | | Transfer | | |
|---|---|---|---|---|---|---|---|
| | DFC [16] | DexGraspNet [17] | ContactGen [35] | UGG [62] | Tink [69] | Ours-Contact | Ours |
| SR (%) ↑ | 78.98 | 83.63 | 73.00 | 70.50 | 69.82 | 78.46 | **84.65** |
| Pen. (mm) ↓ | 3.15 | 4.52 | 4.11 | 8.08 | **1.14** | 1.87 | 1.47 |
| Cov. (%) ↑ | 32.28 | 31.87 | 34.78 | 35.06 | 25.13 | 36.28 | **38.16** |

Table 1: Performance comparison with different task-agnostic grasp generation methods. The best results are in **bold**. *Ours-Contact* denotes the result with only the transferred contact map, while *Ours* denotes the result after using the jointly transferred contact, part and direction maps.

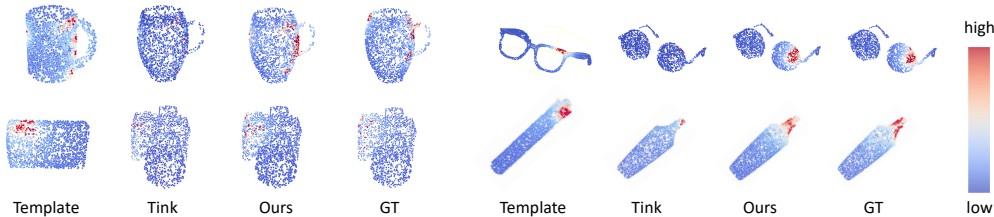

Figure 4: Qualitative comparison with Tink for contact map transfer on novel objects.

**Grasp Synthesis.** Based on the remained points with reliable map values, we exploit an efficient optimization method to recover the grasp parameters. The hand mesh can be reconstructed from $\mathbf{q}$ using differentiable forward kinematic. Given the contact map $\mathbf{c} \in \mathbb{R}^{m \times 1}$, part map $\mathbf{p} \in \mathbb{R}^{m \times b}$, and direction map $\mathbf{d} \in \mathbb{R}^{m \times 3}$, we iteratively update $\mathbf{q}$ by minimizing the following loss function:

$$\mathcal{L}_{\text{syn}} = \lambda_{\text{cont}} E_{\text{cont}} + \lambda_{\text{dir}} E_{\text{dir}} + \lambda_h E_h, \tag{9}$$

where $E_{\text{cont}} = \mathcal{L}_{\text{MSE}}(\hat{\mathbf{c}}, \bar{\mathbf{c}})$ is the mean squared error between the predicted contact map $\hat{\mathbf{c}}$ and the contact map $\bar{\mathbf{c}}$ derived from the current robot configuration $\mathbf{q}$, and $E_{\text{dir}} = \mathbf{w} \cdot \mathcal{L}_{\text{cos}}(\hat{\mathbf{d}}, \bar{\mathbf{d}})$ is the weighted cosine similarity loss. In addition, $E_h$ is a regularization term that ensures grasp quality by penalizing hand-object penetration and encouraging natural hand postures. We optimize 200 steps to obtain the final grasp parameters. Please refer to Sec. A.2 for more details on grasp synthesis.

## 4 Experiment

In this section, we will answer the following key questions through our extensive experiments. (1) Does our transfer-based framework enable high-quality grasp transfer to novel objects, and how does it compare to existing grasp generation methods? (2) How well does our method generalize to novel objects, unseen object categories, and new task specifications, compared to state-of-the-art task-oriented grasp generation approaches? (3) How does each proposed module contribute to the overall performance, including the adaptation module for conditional diffusion, cascaded diffusion framework, and robust grasp optimization method?

### 4.1 Experimental Settings

**Datasets.** To comprehensively evaluate dexterous grasp generation methods across various objects and robotic manipulation tasks, we conducted experiments on the customized CapGrasp dataset [31]. CapGrasp is one of the largest publicly available task-oriented grasp datasets, comprising 1.8k object instances from 32 diverse categories and providing high-quality human grasps for 50k tasks, each with specific textual descriptions. To assess grasp generation performance in robotic manipulation tasks, we first applied the grasp retargeting method from [32] to convert human grasps into ShadowHand [72] grasps. Next, we excluded object categories with only a few instances (*e.g.*, those containing a single instance) and used the remaining categories for model training and evaluation. Specifically, we selected 16 out of the remaining 24 object categories for model training. Within each training category, approximately $10\%$ of the objects were held out for testing to evaluate the model's ability to generalize to novel objects. Meanwhile, the remaining 8 categories were reserved to assess the model's generalization to novel object categories.

**Competing Methods and Evaluation Metrics.** To answer question (1), we compared our method with representative dexterous grasp generation methods, including conventional analytical meth-

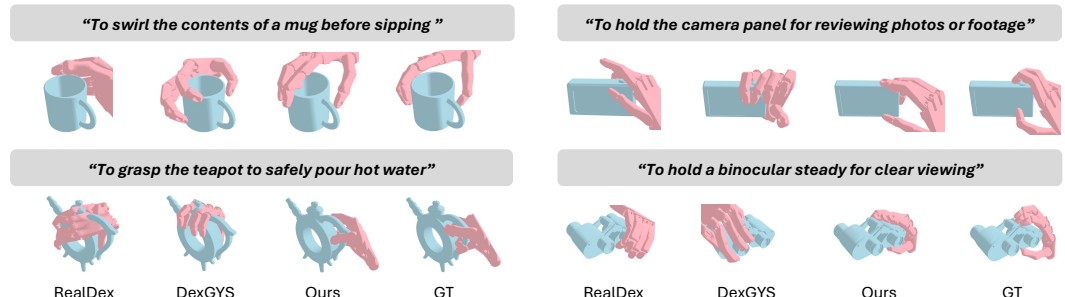

Figure 5: Qualitative comparison with generative methods on unseen objects across diverse tasks.

| Methods | Seen Categories | | | | | Unseen Categories | | | | |
|---|---|---|---|---|---|---|---|---|---|---|
| | SR ↑ | Pen. ↓ | Cov. ↑ | Cont. Err. ↓ | Consis. ↑ | SR ↑ | Pen. ↓ | Cov. ↑ | Cont. Err. ↓ | Consis. ↑ |
| RealDex [50] | 42.16 | 3.26 | 23.61 | 0.1002 | 80.62 | 29.90 | 3.35 | 17.15 | 0.0975 | 70.85 |
| DexGYS [32] | 41.56 | 22.25 | 30.50 | 0.0834 | 74.85 | 39.16 | 23.63 | 32.37 | 0.1332 | 68.08 |
| Tink [69] | 62.60 | **1.29** | 24.86 | 0.0327 | 68.75 | — | — | — | — | — |
| Ours-Contact | 76.14 | 1.68 | 33.55 | 0.0322 | 75.26 | 70.34 | 1.59 | **35.66** | 0.0410 | 75.00 |
| Ours | **79.32** | 1.74 | **36.77** | **0.0287** | **83.60** | **74.14** | **1.36** | 30.05 | **0.0363** | **79.28** |

Table 2: Performance comparison on Seen and Unseen Categories across different methods. Cont. and Consis. are two metrics used to evaluate the alignment of the generated grasp with the task specifications. The best results are in **bold**.

ods DFC [16] and DexGraspNet [17], generative methods ContactGen [35] and UGG (a recent diffusion-based method) [62], and one transfer-based method Tink [69]. Following [32, 31, 34], we quantitatively evaluate the grasp quality using representative metrics, including the overall grasp Success Rate (SR) in IsaacGym simulator [73], Penetration Depth (Pen.), and Contact Coverage (Cov.). To answer question (2), we further compared our method with two recent generative approaches, RealDex [50] and DexGYS [32], assessing their performance on novel object instances and categories across various tasks. In this experiment, we additionally assess the alignment of the generated grasps with the task specifications using representative metrics [74, 31, 32], including Contact Error (Cont. Err.), VLM-assisted [75] Consistency (Consis.), Chamfer Distance (CD), R-Precision (R-Prec@TopK), and Fréchet Inception Distance (P-FID). Please refer to Sec. A.3 and Sec. A.4 for more implementation details.

## 4.2 Main Results

**Quality evaluation in task-agnostic grasping.** Focusing on evaluating the grasp quality, similar to [22, 23, 35], we evaluated different methods on novel objects belonging to seen categories. For each object category, we select five task-agnostic grasps with the lowest retargeting loss from its corresponding shape template, and transfer them to various novel objects within the same category using our proposed diffusion model. For analytical methods, we perform individual analytical optimization with force closure for each novel object and select the five grasps with the lowest analytical energy function values for evaluation. For generative methods, we follow the pipeline in [44] to select the top five grasps from the generated grasp candidates for quantitative assessment.

Tab. 1 presents the average performance of all competing methods. As can be observed, both our method and the analytical methods achieve a high grasp success rate across various objects. It indicates that our method can effectively transfer dexterous grasps from a template to diverse novel objects, maintaining high grasp quality even without performing complex force-closure-based optimization for each novel object. Compared to existing generative approaches, our method consistently produces higher-quality dexterous grasps for various objects, achieving a higher success rate, greater stability with smaller penetration, and better hand coverage over the object. We also compared our method with Tink, a widely used grasp transfer approach. Tink requires learning an implicit shape function for each object instance and performs grasp transfer through shape interpolation. To ensure a fair comparison, we retrieved the most similar shape from the training set for each novel object based on Chamfer Distance and use its corresponding implicit shape for grasp transfer in Tink. However, Tink

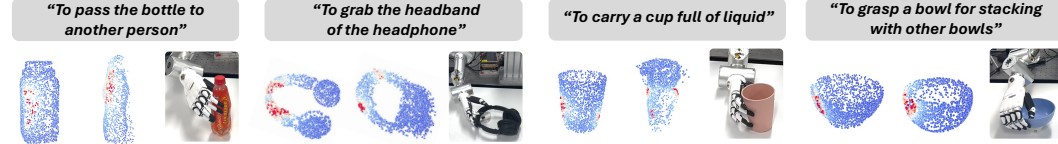

"To pass the bottle to another person"  "To grab the headband of the headphone"  "To carry a cup full of liquid"  "To grasp a bowl for stacking with other bowls"

Figure 6: Real-world experimental results. For each task, the left, middle, and right images show the template contact map, the transferred contact map, and the generated dexterous grasp, respectively.

| Methods | SR | Pen. | Cov. | Cont.Err. |
|---|---|---|---|---|
| w/o $\mathcal{A}_a$ | 37.57 | 1.95 | 21.11 | 0.0522 |
| w/o $\mathcal{A}_e$ | 38.03 | 1.99 | 21.91 | 0.0512 |
| w/o cascaded-I | 60.71 | 1.87 | 25.08 | 0.0489 |
| w/o cascaded-II | 58.51 | 1.82 | 20.73 | 0.0594 |
| w/o task desc. | 73.85 | 1.83 | 34.03 | 0.0322 |
| w/o robust opt. | 55.04 | **1.74** | 27.95 | 0.0448 |
| Ours | **79.32** | **1.74** | **36.77** | **0.0287** |

Table 3: Quantitative ablation study results.

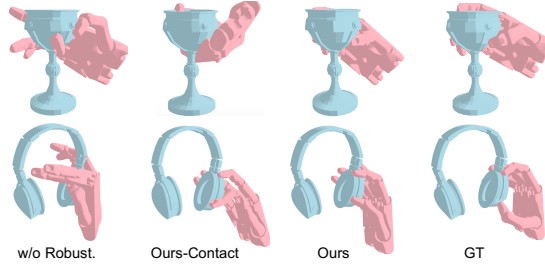

w/o Robust.    Ours-Contact    Ours    GT

Figure 7: Qualitative grasp comparison with / without robust optimization.

is highly sensitive to shape variations, leading to a noticeable drop in grasp quality when transferring grasps to novel objects (as shown in Fig. 4). These results demonstrate that our proposed method can effectively transfer grasps from shape templates to a wide range of novel objects, achieving grasp quality comparable to conventional analytical methods. Meanwhile, our method achieves significantly higher grasp quality compared to existing generative approaches and greater robustness to shape variations compared to Tink.

**Generalization evaluation in task-oriented grasping.** In this experiment, we compared our approach with two recent task-oriented dexterous grasp generation methods, RealDex and DexGYS, as well as the transfer-based method Tink. For RealDex, we first generated a set of grasp candidates and then use a vision-language model prompted with the task description to select the grasp that best aligns with the given task. For DexGYS, the task description is directly used as an input feature to generate a dexterous grasp that aligns with the specified task. Tab. 2 presents a comparative analysis of different methods on novel objects and novel object categories. As can be observed, our method effectively transfers grasps from shape templates to diverse novel objects while maintaining high grasp quality and strong alignment with task specifications. As shown in Fig. 5, across various novel objects, our approach significantly outperforms all competing methods. For the more challenging scenario of novel object categories with novel tasks, all methods experience a performance drop to varying degrees. However, our proposed method demonstrates superior generalization ability. Without requiring retraining on new object categories, it can be directly applied to novel categories to generate stable dexterous grasps for a wide range of manipulation tasks. These results indicate the superior generalization capability of the proposed method for dexterous grasp generation. Please refer to Sec. A.5 for more results.

**Real world experiments.** We conducted real-world dexterous grasping experiments to evaluate the effectiveness of our method in practical scenarios. The experiments were performed on a self-developed humanoid robotic platform equipped with an Inspire dexterous hand mounted at the end of the robot arm. For visual perception, we deployed one ZED camera on the robot's head and placed two RealSense cameras on the left and right sides of the robot to capture multi-view RGB images. Given a textual task instruction, we used Grounding-DINO [76] to segment the target object and leveraged the recent 3D foundation model VGGT [77] to reconstruct the object's point cloud from the captured views. Without any re-training or fine-tuning of our model, we directly used the reconstructed point cloud to predict the dexterous grasp configuration. For grasp execution, we followed the protocol in [32], where the robot arm first moves to the predicted 6-DOF pose of the hand's root, and then the Inspire hand actuates its joint angles based on the predicted contact poses. We tested four object categories across five different task specifications, and repeated the grasping process five times and recorded the average grasp success rate for each object. A successful grasp

is defined as lifting the object at least 30 cm from the table and holding it stably for 5 seconds. As shown in Fig. 6, our method can effectively transfer the contact map from template objects to novel, noisy objects, and achieve an average success rate of 70% across the tested categories (60% for Bowl, 80% for Bottle, 80% for Cup, and 60% for Headphones), demonstrating the practical applicability of our approach and its robustness to real-world observation noise.

### 4.3 Ablation Study

In the ablation study, we removed different individual modules from the complete diffusion model and retrained the model with the same training dataset. We then evaluated the dexterous grasp generated by different model variants on novel objects and Tab. 3 presents the experimental results.

**(i)** Removing the adaptation module would result in a significant performance drop. Both adaptation modules play important roles in the dual-branch diffusion architecture. They improve the learning of complex geometric features for the shape template and novel objects, and effectively model the shape similarity between the shape template and novel objects for robust contact map transfer.

**(ii)** Our proposed cascaded framework is effective in improving grasp generation performance. We validate this through two ablations: (I) using three independent diffusion models separately for contact, part and direction map, and (II) a unified diffusion model that concatenates all three maps into a single representation. The independent variant suffers from inter-map inconsistencies, while the unified model fails to disentangle competing learning signals (please refer to Sec. A.5 for more results). This demonstrates that the cascaded framework ensures better coherence among different grasp representations, ultimately improving grasp generation accuracy.

**(iii)** Incorporating textual task descriptions further enhances grasp generation. This is because the textual features improve the accuracy of the extracted conditioning features. For the same template-object pair, different textual inputs produce different condition features, which guide the model to generate task-specific map predictions. This confirms that task descriptions play a crucial role in adapting the generated grasps to different manipulation tasks.

**(iv)** Without the proposed robust optimization strategy, we observe that the grasp generation performance is even worse than using only the contact map (55.04 vs. 76.14). This is because, while the part map and direction map offer more detailed hand-object contact relationships, their generated results are also more susceptible to noise. As shown in Fig. 7, our proposed robust optimization strategy effectively filters out unreliable predictions in the part map and direction map, ensuring that useful information from these additional maps enhances grasp precision. Moreover, it significantly improves the robustness of the optimized grasp parameters, leading to more reliable grasp generation.

## 5 Conclusion

In this paper, we propose a novel transfer-based framework for dexterous grasp generation, integrating the strengths of both analytical and generative methods. We introduce a conditional diffusion model that leverages task embeddings to learn geometric similarities for contact map transfer, and further develop a cascaded diffusion framework to jointly transfer contact, part, and direction maps while maintaining their consistency. To enhance robustness, we propose an adaptive optimization strategy for reliable grasp parameter optimization. Extensive experiments show our method delivers high grasp quality and task adaptability, with strong generalization to unseen objects and categories, highlighting its potential for dexterous robotic and humanoid applications.

While our method shows promising results, our study is currently limited to experiments conducted on one of the most representative five-fingered robotic hands, the ShadowHand. To further evaluate the generality of our approach, we have conducted additional experiments on human hand grasping. The results demonstrate that our framework can be directly applied to human hands and achieves superior performance in both task-agnostic and task-oriented scenarios (please refer to Sec. A.5 for detailed results). Nevertheless, further exploration of the framework's generalization to cross-embodiment dexterous hands with varying numbers of fingers and morphological structures remains an important and promising direction for future research. Moreover, investigating effective execution policies for dexterous grasping across diverse objects, as well as exploring how the transferred contact map can serve as an enhanced visual representation for vision-language-action models, constitutes another interesting and challenging problem for real-world dexterous manipulation.

**Acknowledgments** This research work has been supported by grants from the National Natural Science Foundation of China (Project No. 62322318), the Research Grants Council of Hong Kong, China (Project No. C4042-23GF), and InnoHK of the Government of Hong Kong via the Hong Kong Centre for Logistics Robotics.

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

# A Appendix

In this supplementary material, we provide additional contents that are not included in the main paper due to the space limit:

## A.1 Details on Cascaded Diffusion Models

**Diffusion Model for Part Map Transfer.** As mentioned in the main paper, the generation of target's part map $\mathbf{p}_a$ is conditioned on $(\mathbf{x}_e, \mathbf{c}_e, \mathbf{p}_e, \mathbf{x}_a, \hat{\mathbf{c}}_a)$. The diffusion loss for the part map transfer is defined as:

$$\mathcal{L}_{\text{diff}}^{\text{part}} = \mathbb{E}_{t \sim \mathcal{U}(1,T), \epsilon \sim \mathcal{N}(0,\mathbf{I})} \left\| \epsilon - g_{\text{dec}}^{\text{part}}(\mathcal{A}_a(\mathbf{h}_a^t, \mathbf{h}_e), t) \right\|^2 ,$$

where $g_{\text{dec}}^{\text{part}}$ is the decoder for the part map diffusion process. For notational consistency with the contact diffusion formulation in the main paper, we reuse $g_{\text{dec}}$ to denote the decoder for the part map diffusion process in this section, and reuse $\mathbf{h}_e$ to denote the template feature (now encoding part information), and $\mathbf{h}_a^t$ for the target feature at timestep $t$. The target feature $\mathbf{h}_a^t$ is obtained by encoding the concatenation of target's point cloud $\mathbf{x}_a$, the predicted contact map $\hat{\mathbf{c}}_a$, and the noisy part map $\mathbf{p}_a^t$ at timestep $t$ using the target encoder $g_{\text{enc}}$:

$$\mathbf{h}_a^t = g_{\text{enc}}(\mathbf{x}_a, \hat{\mathbf{c}}_a, \mathbf{p}_a^t).$$

Similarly, the template feature $\mathbf{h}_e$ is obtained by encoding the template's point cloud $\mathbf{x}_e$, contact map $\mathbf{c}_e$, and part map $\mathbf{p}_e$ using the template encoder $f_{\text{enc}}$:

$$\mathbf{h}_e = f_{\text{enc}}(\mathbf{x}_e, \mathbf{c}_e, \mathbf{p}_e).$$

These features are then adapted through the adaptation module $\mathcal{A}_a$, which integrates $\mathbf{h}_e$ into the target branch to guide the generation of the target part map. The overall loss function for part map generation combines the reconstruction loss and the diffusion loss:

$$\mathcal{L}_{\text{part}} = \mathcal{L}_{\text{recon}}^{\text{part}} + \lambda_p \mathcal{L}_{\text{diff}}^{\text{part}},$$

where $\lambda_p = 1$ is the weighting constant.

During inference, we randomly sample a noise $\mathbf{z}_T \sim \mathcal{N}(0, \mathbf{I})$ and perform $T$ denoising steps to gradually obtain the predicted part map $\hat{\mathbf{p}}_a \triangleq \mathbf{z}_0$ conditioned on $\mathbf{x}_e$, $\mathbf{x}_a$, $\mathbf{c}_e$, $\mathbf{p}_e$, $\hat{\mathbf{c}}_a$, and $t$.

**Diffusion Model for Direction Map Transfer.** Similarly, the diffusion loss for the direction map transfer is defined as:

$$\mathcal{L}_{\text{diff}}^{\text{dir}} = \mathbb{E}_{t \sim \mathcal{U}(1,T), \epsilon \sim \mathcal{N}(0,\mathbf{I})} \left\| \epsilon - g_{\text{dec}}^{\text{dir}}(\mathcal{A}_a(\mathbf{h}_a^t, \mathbf{h}_e), t) \right\|^2 ,$$

where $g_{\text{dec}}^{\text{dir}}$ is the decoder for the direction map diffusion process, $\mathbf{h}_a^t$ is the target feature at timestep $t$, and $\mathbf{h}_e$ is the template feature.

The target feature $\mathbf{h}_a^t$ is obtained by encoding the target's point cloud $\mathbf{x}_a$, the predicted contact map $\hat{\mathbf{c}}_a$, the predicted part map $\hat{\mathbf{p}}_a$, and the noisy direction map $\mathbf{d}_a^t$ at timestep $t$ using the target branch encoder $g_{\text{enc}}$:

$$\mathbf{h}_a^t = g_{\text{enc}}(\mathbf{x}_a, \hat{\mathbf{c}}_a, \hat{\mathbf{p}}_a, \mathbf{d}_a^t).$$

The template feature $\mathbf{h}_e$ is obtained by encoding the template's point cloud $\mathbf{x}_e$, contact map $\mathbf{c}_e$, part map $\mathbf{p}_e$, and direction map $\mathbf{d}_e$ using the template branch encoder $f_{\text{enc}}$:

$$\mathbf{h}_e = f_{\text{enc}}(\mathbf{x}_e, \mathbf{c}_e, \mathbf{p}_e, \mathbf{d}_e).$$

The overall loss function for direction map generation combines the reconstruction loss and the diffusion loss:

$$\mathcal{L}_{\text{dir}} = \mathcal{L}_{\text{recon}}^{\text{dir}} + \lambda_d \mathcal{L}_{\text{diff}}^{\text{dir}},$$

where $\lambda_d = 1$ is the weighting constant.

## A.2 Details of Grasp Synthesis

The regularization term $E_h$ ensures grasp quality by incorporating penetration, naturalness, and hand-object distance into the optimization. It is defined as:

$$E_h = \lambda_{\text{pen}} E_{\text{pen}} + \lambda_{\text{q}} E_{\text{q}} + \lambda_{\text{dis}} E_{\text{dis}}, \tag{10}$$

where $\lambda_{\text{pen}}$, $\lambda_q$, and $\lambda_{\text{dis}}$ are weighting coefficients that balance the contributions of each term. In detail, the penetration penalty $E_{\text{pen}}$ is computed as:

$$E_{\text{pen}} = \sum_{h \in \mathcal{H}} \text{ReLU}(-\varsigma(h, \mathbf{x})), \tag{11}$$

where $\varsigma(h, \mathbf{x})$ is the signed distance from a point $h$ sampled from the hand mesh $\mathcal{H}$ to the object $\mathbf{x}$. The naturalness term $E_q$ penalizes deviations of joint angles $\theta$ from their upper and lower limit $\theta_{\text{upper}}$ and $\theta_{\text{lower}}$:

$$E_q = ||\text{ReLU}(\theta - \theta_{\text{upper}}) + \text{ReLU}(\theta_{\text{lower}} - \theta)||^2. \tag{12}$$

Finally, the distance term $E_{\text{dis}} = d(h_k, \mathbf{x})$ encourages the predefined keypoints $h_k$ to be closer to the object surface. Here, $d(\cdot, \cdot)$ computes the Euclidean distance between the points and object.

## A.3 Implementation Details

We set the number of points $n$ and $m$ for the template and target point cloud to 2048, respectively. We employ PointNet++ [78] as the backbone for both the template reconstruction model and the target diffusion model, which contains 4 Set Abstraction layers and 4 Feature Propagation layers as point encoder and point decoder following the standard U-Net [79] framework. The feature dimensions for $\mathbf{h_e}$ and $\mathbf{h_a}$ are both 512. The diffusion timestep $t$ is embedded into feature with 512 dimensions via a two-layer MLP ($\mathbb{R}^{128} \rightarrow \mathbb{R}^{512} \rightarrow \mathbb{R}^{512}$) with Swish activation. We use a pre-trained language foundation model, Bert-base [71], to extract the token embedding $f_l(\ell) \in \mathbb{R}^{768}$ from the task description. The adaption module $\mathcal{A}$ contains 4 attention heads and 64 hidden dimensions. We employ the Adam optimizer with a batch size of 56, number of workers 4, and learning rate 2e-4, jointly training the two branches from scratch for 1000 epochs on two NVIDIA 3090 GPUs, which takes about 24 hours. The diffusion timesteps $T$ are set to 1000 for both training and sampling. For robust grasp optimization, the thresholds $\tau_a$ and $\tau_b$ are both set to 0.1. The weighting coefficients for grasp synthesis are configured as $\lambda_{cont} = 1 \times 10^{-1}$, $\lambda_{dir} = 1 \times 10^{-2}$, $\lambda_h = 1.0$, $\lambda_{pen} = 3.0$, $\lambda_q = 1.0$, and $\lambda_{kp} = 1.0$.

## A.4 Details on Experimental Settings

Our customized CapGrasp [31] dataset contains 24 categories, divided into seen and unseen sets with strict category-level separation to evaluate generalization:

- **Seen Categories** (16 categories):
  Binoculars, Bottle, Cameras, Cylinder bottle, Eyeglasses, Frying pan, Hammer, Light bulb, Lotion pump, Mouse, Mug, Pen, Phone, Power drill, Screwdriver, and Teapot.

- **Unseen Categories** (8 categories):
  Bowl, Cup, Flashlight, Game controller, Headphones, Knife, Trigger sprayer, and Wineglass.

The dataset consists about 50k hand-object grasp pairs, each annotated with on average 5 task descriptions. During training, we randomly select one task description for each grasp. To facilitate training efficiency and balance the data distribution, for object categories containing more than 2k grasp pairs, we randomly retain at most 2k pairs per epoch during training. This sampling strategy is applied consistently to both our proposed method and all baseline methods to ensure fair comparison.

As discussed in the main paper, we evaluate the quality of generated grasps using Success Rate, Penetration Depth, and Contact Coverage. **Success Rate (SR)** is evaluated in the IsaacGym simulator [73]. A successful grasp involves lifting an object with gravity at $-9.8 m/s^2$, raising it over 10 cm, and ensuring that the object displacement remains below 2 cm after 60 simulation steps. We randomize the object position and rotation on table and run 10 trials for each grasp, taking the average as the success rate. To ensure a secure grasp in the simulator, we follow [22] and apply force through single-step pose optimization for all compared methods. Initially, we select finger points facing the

Table 4: Comparison of the generated human grasps on unseen objects.

| Method | Pen. Volume ↓ | Contact Ratio ↑ | Simulation Disp. ↓ |
|---|---|---|---|
| ContactGen [35] | 5.44 | 82.75% | 3.96 |
| Tink [69] | 1.91 | 77.62% | 3.31 |
| Ours | **1.11** | **87.45%** | **2.76** |

object and within 5mm of its surface, then define a target hand pose by moving these points 2mm toward the object. Next, we optimize the hand joint parameters using a single step of gradient descent to minimize the discrepancy between the current and target poses. Finally, the force is applied to the object by updating the hand pose via a positional controller. **Penetration Depth (Pen.)** measures the maximum penetration depth from the object point cloud to the hand meshes, quantifying surface penetration. **Contact Coverage (Cov.)** computes the percentage of hand points within ±2mm of the object surface, reflecting the ratio of points in contact with the object.

Following [31, 32, 74], we adopt Contact Error, P-FID, Chamfer Distance, R-Precision and VLM-assisted approach to evaluate the task consistency. **Contact Error (Cont. Err.)** calculates the L2 distance of object contact map between the generated grasp and ground truth. **P-FID** computes the Fréchet Inception Distance between the generated hand point cloud and the ground truth point clouds, leveraging a pre-trained feature extractor from [80]. **Chamfer Distance (CD)** quantifies the average point-wise discrepancy between the generated hand point cloud and the ground-truth point cloud. **R-Precision (R-Prec@TopK)** evaluates the semantic alignment between the language instructions and the generated grasps. We render the generated hand-object interactions into images and use pre-trained image and text encoders from [80] to measure how well the visual output matches the textual description. For each generated grasp image, we create a candidate pool composed of its ground-truth instruction and 31 randomly selected samples. We then compute the cosine distances between the image features and the language features for every sample in the pool and then rank them accordingly. We report the Top-k retrieval accuracy (for k=1, 3, 5), which is the percentage of trials where the ground-truth instruction is successfully ranked among the top k candidates. **VLM-assisted Consistency (Consis.)** is performed by prompting the GPT-4o [75] to score the consistency of the synthesized grasp based on the grasp images and task instructions. The scores provided range from 0 to 100, with higher scores indicating better consistency.

## A.5  Additional Results

**Evaluation on Human Grasp Generation.** In the main paper, we primarily use robotic hand [72] for evaluation. Apart from the robotic hand, we also evaluate our method on the human grasp generation. In specific, we change the three object-centric maps on template from robotic hand to human hand parameterized in MANO model [81]. We compare our method with one generative method, ContactGen [35], and one transfer-based method, Tink [69]. Both methods leverage contact map, part map, and direction map as intermediate representations for dexterous grasp generation. Following [35], we evaluate the generated grasps based on the Penetration Volume (Pen. Volume), Contact Ratio, and Simulation Displacement of the object center under gravity (Simulation Disp.). As shown in Table 4, our method achieves the best performance in all metrics, indicating that our method can also be effectively applied to MANO model. Fig. 16 shows the visualization results of our method on unseen objects and unseen categories for various task descriptions.

**More Visualization Results.** Fig. 13 shows more qualitative results generated by our method. Fig. 10 presents a comparison with Tink [69] for contact map transfer on novel objects. Fig. 15 further demonstrates the visualization results of our proposed method performing contact transfer on novel object categories. The model effectively captures the geometric relationships between novel template and target objects, demonstrating strong generalizability across diverse and challenging scenarios.

**Additional Results on Generalization Evaluation.** Fig. 11 provides a qualitative comparison with RealDex [50] and DexGYS [32] on novel objects across various tasks. Additionally, Fig. 14 illustrates the grasps generated by our proposed method on unseen categories. Tab. 5 shows the quantitative comparison of generative metrics among our method, RealDex, and DexGYS. Our method significantly outperforms other methods, showing superior generalization ability across a wide range of tasks and unseen categories. To further enhance the comparative analysis with generative methods for grasp transfer, we also modified the baseline DexGYS to incorporate template

Table 5: Quantitative comparison of generative metrics with generative methods on novel categories.

| Method | P-FID↓ | CD↓ | R-Prec@Top1↑ | R-Prec@Top3↑ | R-Prec@Top5↑ |
|---|---|---|---|---|---|
| RealDex | 14.564 | 0.155 | 0.1081 | 0.1662 | 0.3784 |
| DexGYS | 10.344 | 0.120 | 0.1351 | 0.3243 | 0.4595 |
| Ours-contact | 6.381 | 0.037 | **0.2569** | 0.4538 | 0.5972 |
| Ours | **6.124** | **0.028** | 0.2322 | **0.4795** | **0.6199** |

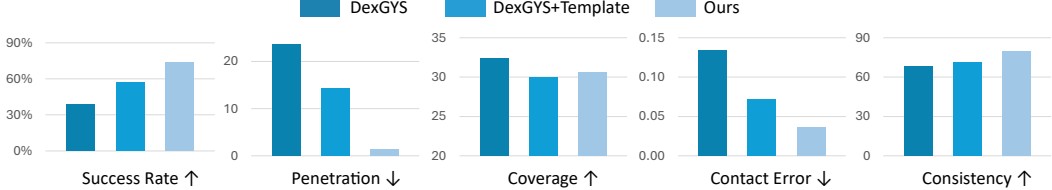

Figure 8: Comparison with generative baseline with or without template on unseen categories

information. Specifically, we introduced a new template branch in the first-stage module of DexGYS, which takes both the template object point cloud and the corresponding hand point cloud as inputs. We extracted template-relevant features using a point cloud backbone and concatenated them with the original target object and textual task features to condition the grasp generation process. We re-trained this customized DexGYS+Template model from scratch on the same training dataset used for our method. As shown in the Fig. 8, incorporating the template grasp improves the performance of DexGYS to some extent, especially in terms of grasp quality and task relevance. However, the performance of this variant remains consistently inferior to our proposed model across all 8 unseen object categories. This result suggests that simply using the template grasp as an additional condition, without explicitly modeling the geometric similarity between the template and the target object, is insufficient for generalizing to novel categories.

**Ablation Study on Cascaded Framework.** Fig. 17 compares the transferred part and direction maps between models trained independently (w/o cascaded-I) and our cascaded framework. The model without the cascaded framework (columns *(b,f)*) exhibits significant inconsistency across the three maps, highlighting the importance of the cascaded design.

**Ablation Study on Task Embedding.** Fig. 12 compares the transferred contact maps and generated grasps between the model without task embedding and our method. Our method (columns *(b,d)*) aligns better with the input task, indicating that incorporating task descriptions into the model enhance grasp adaptation to diverse tasks.

Table 6: Robustness to template selection strategy on novel categories.

| Method | SR↑ | Pen.↓ | Cov.↑ | Cont. Err.↓ | Consis.↑ |
|---|---|---|---|---|---|
| Ours | 74.14 | 1.36 | 30.55 | 0.0363 | 79.28 |
| Ours-Random | 75.36 | 1.50 | 32.32 | 0.0391 | 77.19 |

**Robustness to Template Variations.** In main paper, for each object category, one shape template is randomly selected from the CapGrasp dataset and used consistently across all experiments. To further evaluate our model's sensitivity to the choice of shape template, we conducted additional experiments on 8 unseen object categories using varying template shapes. Specifically, for each target object during testing, we randomly selected a different shape template (from the same category but not the same instance) from CapGrasp, and directly applied our trained model for grasp transfer without any re-training. In addition, we also conducted experiments using alternative template hands generated by two different methods: DexGraspNet (an analytical method) and ContactGen (a generative method). We sampled a diverse set of grasps around the shape template using these methods, then executed them in simulation and selected the top-10 grasps based on success rate as template hands for grasp transfer. We then evaluated the transferred grasps using three label-free quality metrics. As shown in the Tab. 6 and Fig. 9, our model maintains strong performance across different template hand sources,

demonstrating that the proposed grasp transfer framework is not restricted to a fixed shape template and is robust to shape variation between the template and the target object.

## A.6 Discussion on Broader Impacts

In this paper, we propose a transfer-based framework for dexterous grasp generation. By enabling robust grasp transfer across objects within categories, our method can potentially enhance assistive robotics for daily living support, improve industrial automation efficiency in manufacturing and logistics, and facilitate more natural physical interactions in service robotics. The method's ability to handle complex shape variations makes it particularly valuable for applica-

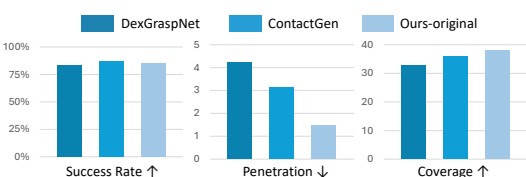

Figure 9: Performance of transferring grasps from different sources

tions requiring adaptable grasping, such as customized prosthetic control, warehouse automation for diverse products, and educational robotics platforms. These advancements may ultimately contribute to increased workplace safety, expanded accessibility for individuals with mobility impairments, and reduced physical strain in repetitive manual tasks.

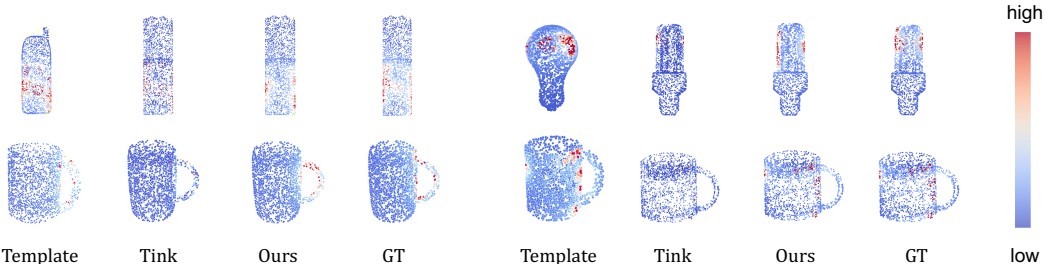

Figure 10: Qualitative comparison results with Tink for contact map transfer on novel objects.

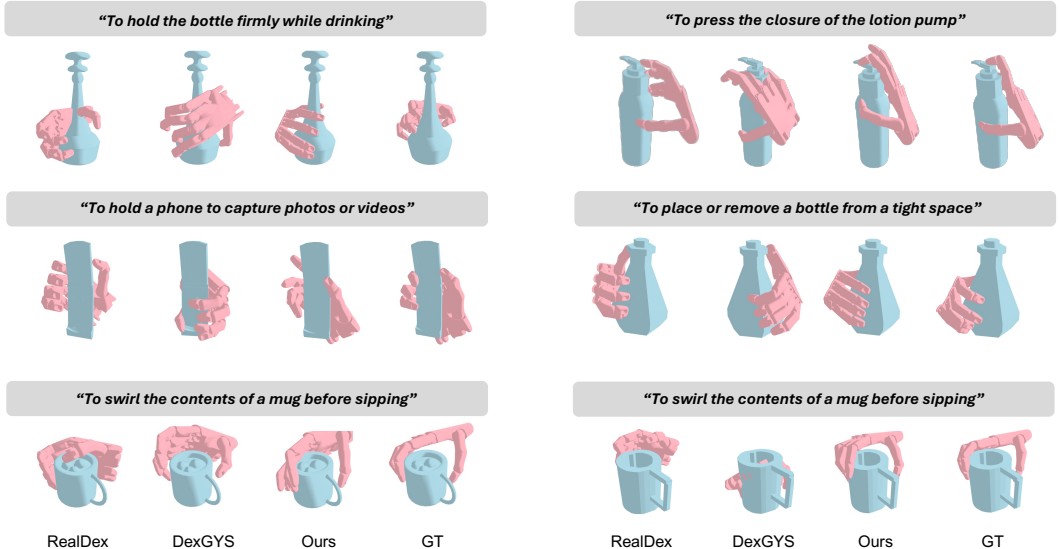

Figure 11: More qualitative comparison with generative methods on unseen objects across diverse task descriptions.

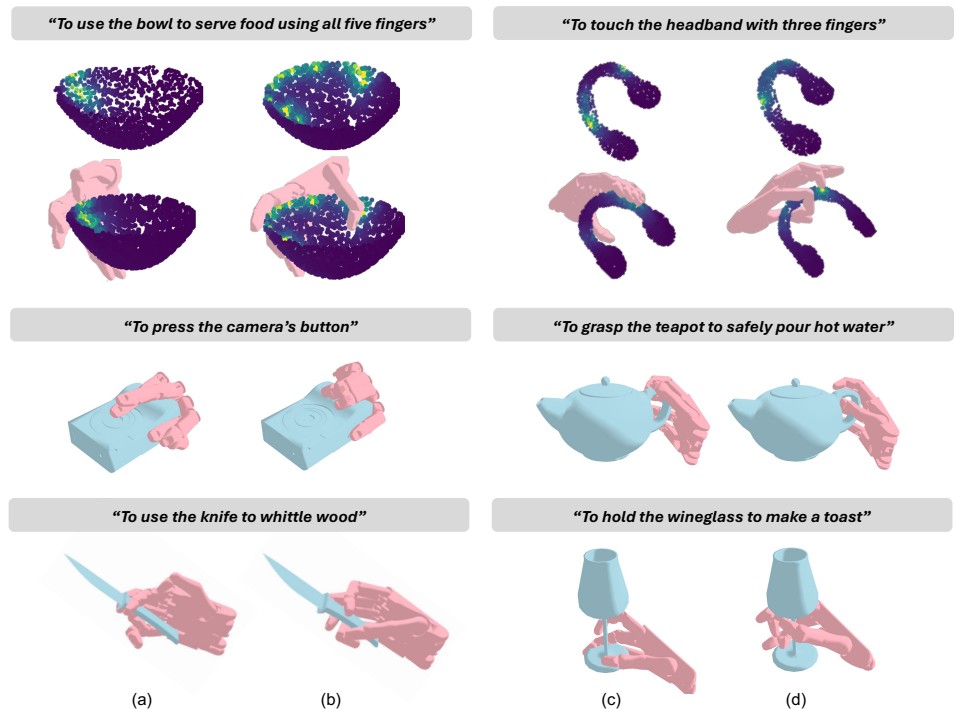

Figure 12: Ablation results on the use of task embedding for contact map transfer and grasp generation. (a,c) Transferred contact maps/grasps from the model without using task embedding. (b,d) Transferred contact maps/grasps from our proposed method.

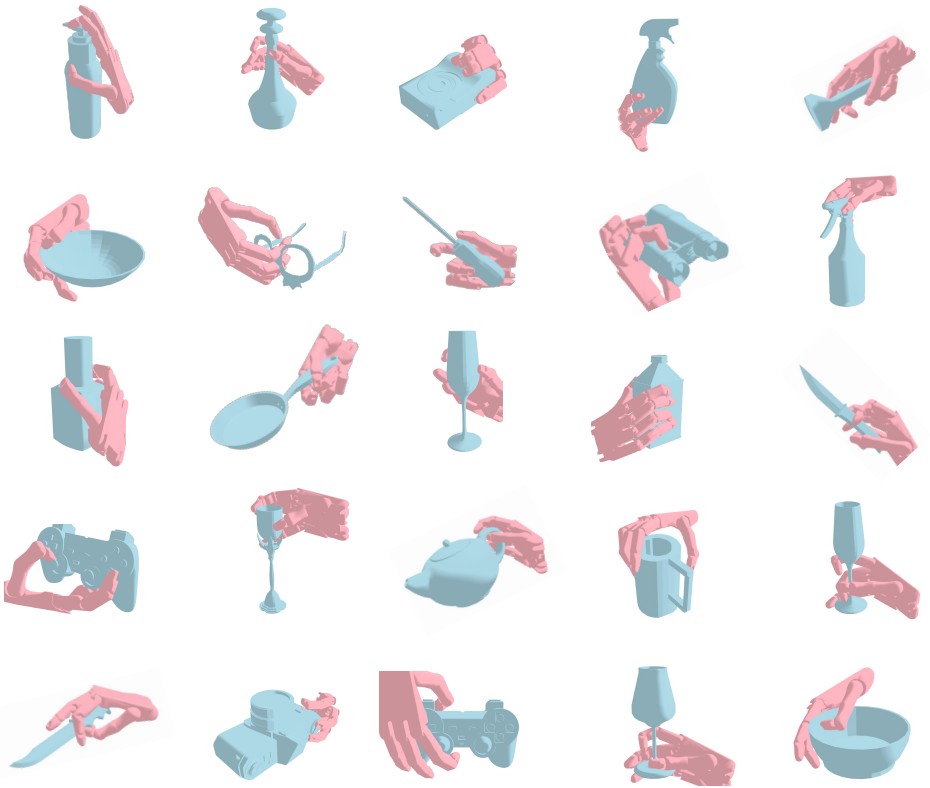

Figure 13: Visualization of grasps generated by our proposed method on diverse objects.

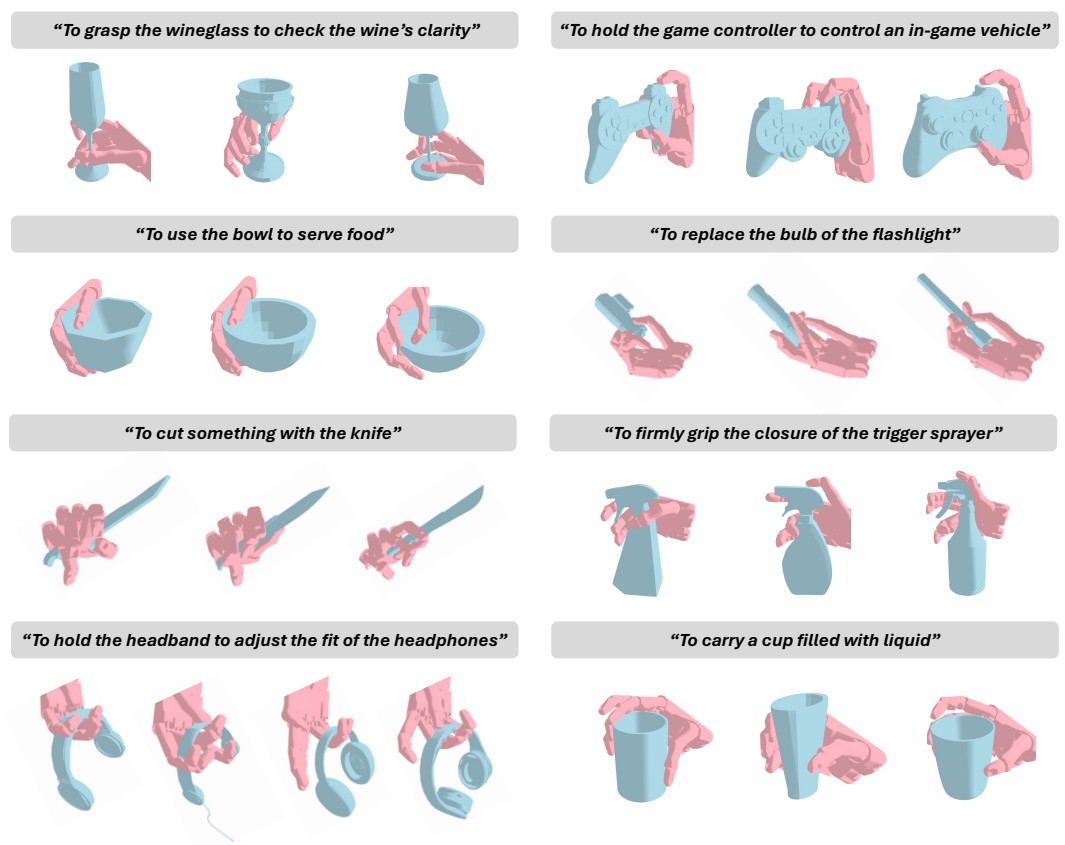

Figure 14: Visualization of the grasps generated by our method on unseen categories for various task descriptions.

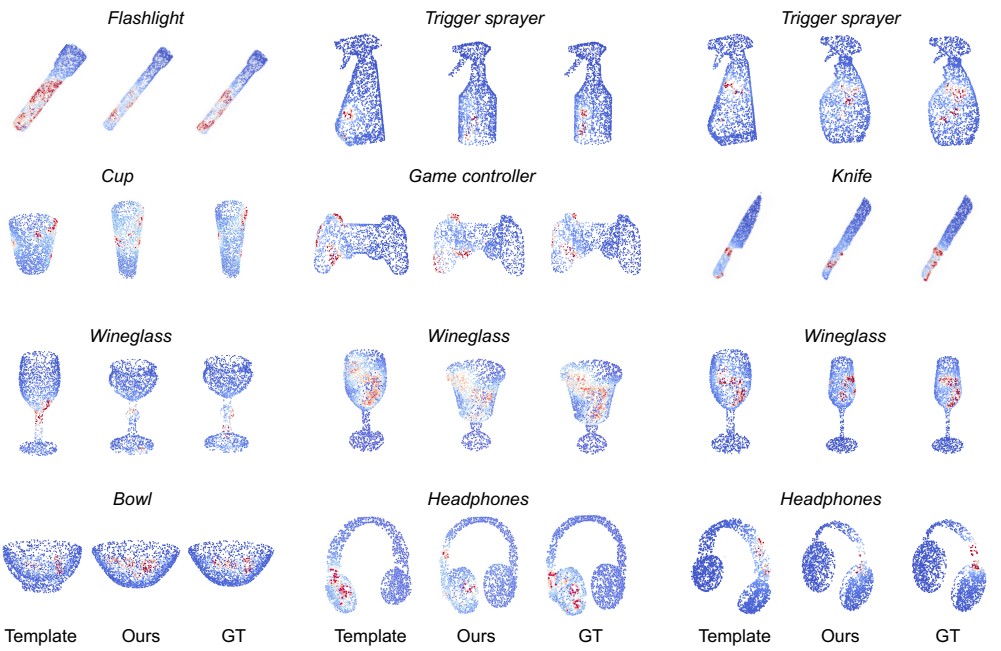

Figure 15: Qualitative results of our method for contact map transfer on unseen categories.

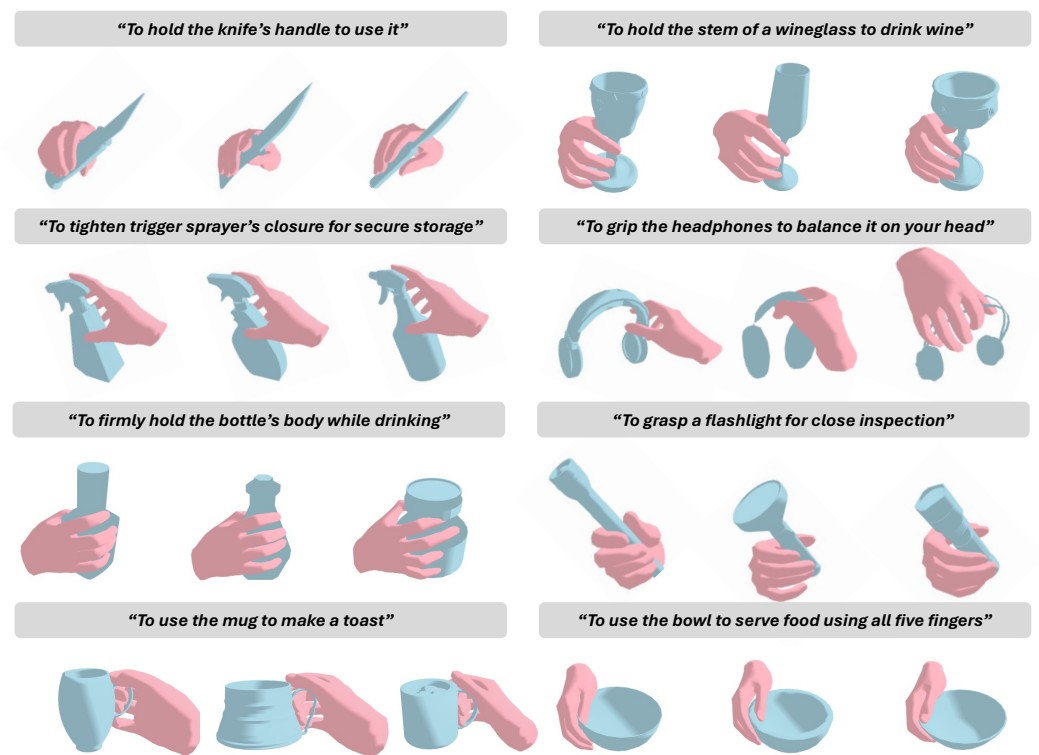

Figure 16: Visualization of the human grasps generated by our method on unseen objects and unseen categories for various task descriptions.

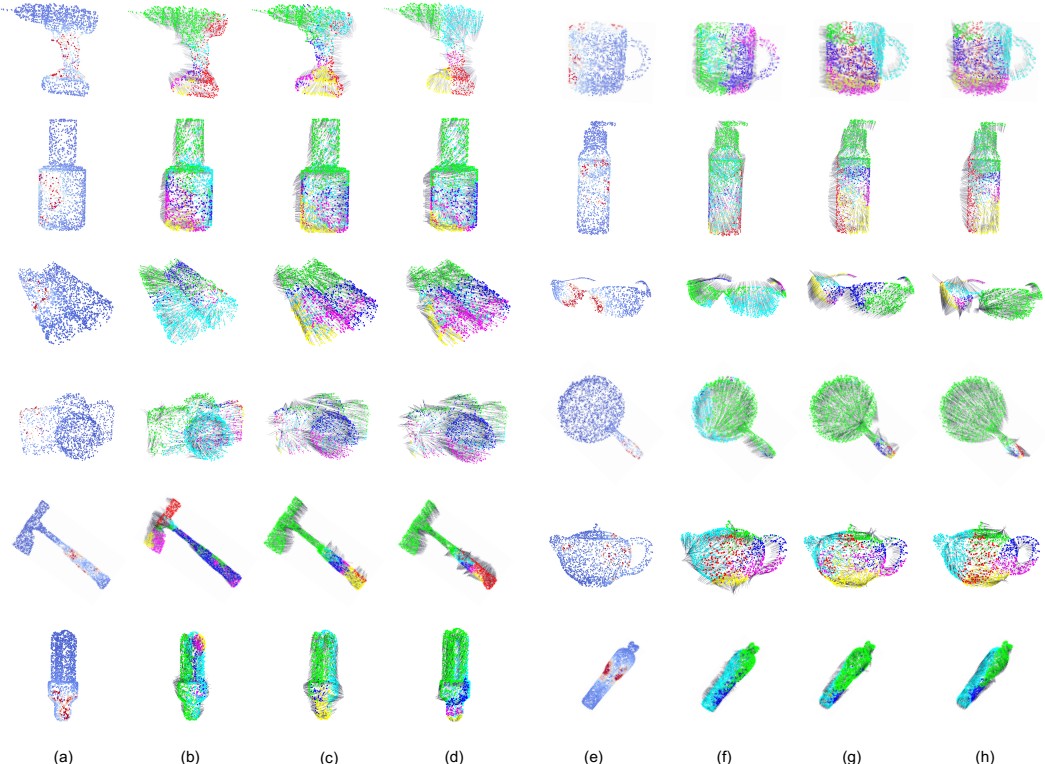

| (a) | (b) | (c) | (d) | (e) | (f) | (g) | (h) |

Figure 17: Qualitative ablation study of the cascaded framework. (a,e) Transferred contact maps. (b,f) Transferred part/direction maps from the model w/o cascaded-I. (c,g) Transferred part/direction maps from our model. (d,h) Ground truth part/direction maps.

