# OpenReview forum: "Contact Map Transfer with Conditional Diffusion Model for Generalizable Dexterous Grasp Generation"
_NeurIPS.cc/2025/Conference — NeurIPS 2025 poster_

### Official Review · Reviewer_jrqf · 2025-07-02

**Clarity:** 3
**Significance:** 3
**Originality:** 3
**Rating:** 4
**Confidence:** 5

**Summary:**

This work proposes a transferable dexterous grasp generation framework based on a conditional diffusion model, aiming to address the inefficiency of analytical methods and the poor generalization of generative approaches. The core idea is to reformulate grasp transfer as object contact generation. It employs a dual-mapping mechanism to handle shape variations, introduces part and direction maps to enhance contact details, and uses a cascaded conditional diffusion model to ensure consistency among them. Finally, a robust grasp recovery mechanism is applied to optimize grasp configurations. Experiments demonstrate that the method excels in grasp quality, efficiency, and generalization, achieving stable transfer to novel objects and tasks and outperforming existing analytical, generative, and transfer-based methods in terms of success rate, penetration depth, and other metrics.

**Questions:**

See Weaknesses

**Ethical Concerns:**

["NO or VERY MINOR ethics concerns only"]

**Final Justification:**

Overall, the strengths of the paper outweigh its weaknesses. It is recommended that the authors update the discussion results from the rebuttal into the final version of the paper.

**Limitations:**

See Weaknesses

**Paper Formatting Concerns:**

No Paper Formatting Concerns

**Quality:**

3

**Strengths And Weaknesses:**

The method is well-motivated, using contact transfer to enable generalization across objects with similar structures. The paper is well-written, with technical contributions clearly and concisely presented.

Weakness
1. The work lacks necessary real-world robot experiments to validate its applicability in practical environments.

2. This paper appears to relay on relatively high-quality point clouds. In real-world settings, point clouds are often incomplete or noisy, which could compromise the effectiveness of the point cloud-based transfer process and introduce uncertainty in the subsequent optimization steps.

3. For dexterous grasp generation, it is encouraged to adopt generative evaluation metrics that better reflect the model’s ability to produce aligned grasps. You may refer to the following papers [1,2].

4. During testing, how the template hand is selected for each sample. And whether this choice can affect model performance.

[1] Li K, Wang J, Yang L, et al. Semgrasp: Semantic grasp generation via language aligned discretization[C]//European Conference on Computer Vision. Cham: Springer Nature Switzerland, 2024: 109-127.

[2] Wei Y L, Lin M, Lin Y, et al. Afforddexgrasp: Open-set language-guided dexterous grasp with generalizable-instructive affordance[J]. arXiv preprint arXiv:2503.07360, 2025.

---

> ### Author Rebuttal · Authors · 2025-07-31
>
> Dear reviewer jrqf, thank you for your thoughtful review and constructive suggestions. In the following, we respond to your questions in detail one by one:
>
> ---
>
> **Q1&Q2: Robustness to object point cloud noise and real-world robot experiments.**
>
> **A1&A2**: We appreciate the reviewer’s important comments. Following your suggestion, we performed additional robustness evaluations in both simulation and real-world settings. It is worth noting that our model was already designed for robustness, having been trained with standard point cloud data augmentation techniques (e.g., random sampling and Gaussian noise). For the new simulation analysis, we followed [A] to randomly mask out 30% of the object points to simulate irregular and incomplete point cloud distributions. As evidenced by the table below, our grasp transfer model's performance remains stable, with no significant degradation in quantitative grasp quality metrics under these conditions.
>
> Table A: Quantitative Results of Robustness to Simulated Point Cloud Noise
>
> | Test Data | SR (%) $\uparrow$ | Pen. $\downarrow$ | Cov. $\uparrow$ | Cont. Err. $\downarrow$ | Consis. $\uparrow$ |
> | --------- | :-----------: | :--------------: | :------------: | :---------------------: | :----------------: |
> | Clean     |     74.14     |       1.36       |     30.05      |         0.0363          |       79.28        |
> | Noisy     |     69.34     |       2.67       |     24.19      |         0.0370          |       74.13        |
>
> Although the grasp success rate saw a minor decrease from 74.14% to 69.34%, we attribute this drop primarily to missing points in critical object regions (such as a mug's handle or a bowl's rim), which hinders accurate contact computation during the final optimization stage. For our real-world experiments, object point clouds were reconstructed from a multi-view camera setup (one ZED, two RealSense). These point clouds naturally contained realistic noise and missing areas owing to partial occlusions and limited sensor coverage. Feeding these imperfect point clouds directly into our grasp transfer pipeline, our method achieved an average success rate of 70% on a dexterous hand, underscoring its strong robustness to real-world artifacts. Due to page constraints, please see our response to Q2 of reviewer UCkK for a more detailed account of these experiments.
>
> [A] Wang, Tao, et al. Sparse convolutional networks for surface reconstruction from noisy point clouds. WACV 2024.
>
> ---
>
> **Q3: More results with generative evaluation metrics.**
>
> **A3**: We thank the reviewer for this valuable suggestion. Following your recommendation, we have adopted three generative evaluation metrics from SemGrasp and Afforddexgrasp to more comprehensively assess the alignment between the generated grasps and their corresponding tasks. ***P-FID*** measures the distributional similarity between generated and ground-truth grasps with the same intention. We leverage the pre-trained feature extractor from [75] for this evaluation. ***Chamfer Distance (CD)*** quantifies the average point-wise discrepancy between the generated hand point cloud and the ground-truth point cloud. ***R-Precision*** evaluates the semantic alignment between the language instructions and the generated grasps. Specifically, we render the generated hand-object interactions into images and use pre-trained image and text encoders from [75] to measure how well the visual output matches the textual description. For each generated grasp image, we create a candidate pool composed of its ground-truth instruction and 31 randomly selected samples. We then compute the cosine distances between the image features and the language features for every sample in the pool and then rank them accordingly. We report the Top-k retrieval accuracy (for k=1, 3, 5), which is the percentage of trials where the ground-truth instruction is successfully ranked among the top k candidates.
>
> We conducted these evaluations on the unseen categories benchmark to assess generalization. As shown in the below table, our method significantly outperforms all baselines across these generative metrics, confirming its superior ability to generalize and maintain semantic consistency.
>
> Table B: Generative Evaluation Metrics on Unseen Categories
>
> | Method       | P-FID $\downarrow$ | CD $\downarrow$ | R-Precision@Top1 $\uparrow$ | R-Precision@Top3 $\uparrow$ | R-Precision@Top5 $\uparrow$ |
> | ------------ | :----------------: | :-------------: | :-------------------------: | :-------------------------: | :-------------------------: |
> | RealDex      |       14.564       |      0.155      |           0.1081            |           0.1662            |           0.3784            |
> | DexGYS       |       10.344       |      0.120      |           0.1351            |           0.3243            |           0.4595            |
> | Ours-contact |       6.381        |      0.037      |         **0.2569**          |           0.4538            |           0.5972            |
> | Ours         |     **6.124**      |    **0.028**    |           0.2322            |         **0.4795**          |         **0.6199**          |
>
> ---
>
> **Q4: How is the template hand selected, and how robust is our model to this choice?**
>
> **A4**: We thank the reviewer for raising this important point. We apologize for the confusion and would like to clarify that in all our main experiments, we consistently use the template hand provided by the CapGrasp dataset. Importantly, for fair comparison, the same template hand is also used by other transfer-based baselines such as Tink. We choose the CapGrasp-defined template hand because it allows us to leverage the provided point-wise ground-truth contact maps on various novel objects, enabling rigorous quantitative evaluation of the transferred contact maps. Additionally, CapGrasp offers ground-truth grasps for different tasks, which further supports task-level evaluation of the transferred grasps using metrics such as Contact Error (Cont. Err.) and Chamfer Distance (CD). However, we emphasize that our method is not limited to this specific template hand. To verify the robustness of our framework to the choice of template hand, we conducted additional experiments using alternative template hands generated by two different methods: DexGraspNet (an analytical method) and ContactGen (a generative method). We sampled a diverse set of grasps around the shape template using these methods, then executed them in simulation and selected the top-10 grasps based on success rate as template hands for grasp transfer. We then evaluated the transferred grasps using three label-free quality metrics.
>
> Table C: Performance of Transferring Grasps from Different Sources
>
> | Source of Template Hand | SR (%) $\uparrow$ | Pen. $\downarrow$ | Cov. $\uparrow$ |
> | ----------------------- | ------------- | ----------------- | --------------- |
> | DexGraspNet             | 82.99         | 4.21              | 32.83           |
> | ContactGen              | **86.88**     | 3.12              | 35.73           |
> | Our original            | 84.65         | **1.47**          | **38.16**       |
>
> The results demonstrate that our model maintains strong performance across different template hand sources, indicating that our framework is robust to the selection and generation process of the template hand. This highlights the generalizability and applicability of our dexterous grasp transfer framework.

---

> > ### Comment · Reviewer_jrqf · 2025-08-04
> >
> > Thank you for the author's reply. Some of my concerns have been addressed, but I still have the following questions:
> >
> > 1. Although Gaussian noise/random sampling/random masking is a general point cloud augmentation technique, it is still difficult to simulate real noise (which comes from multiple sources, such as depth cameras and complete point cloud synthesis).
> >
> > 2. According to the paper, the template hands seem to come from the same category (e.g., from one mug to another, as shown in Figures 2 and 3). So, how should templates be chosen for categories that have never been seen before? The author mentioned 8 unseen categories (Bowl, Cup, Flashlight, Game controller, Headphones, Knife, Trigger sprayer, and Wine glass). Which category do the template hands for these categories come from, and are all the templates derived from a single object example?
> >
> > 3. During inference, how should the template hand and object be selected? Is it predefined?
> >
> > 3. Regarding the concerns about additional results: First, DexGraspNet is a semantic-free force closure synthesis method and requires high-fidelity object models to calculate contact forces and force closure. How can DexGraspNet be integrated into your semantic pipeline? Second, according to [2], the ContactGen method should perform poorly for unseen categories. Why does it achieve performance similar to your method?

---

> > > ### Author Response · Authors · 2025-08-05
> > > **Response to new question 1**
> > >
> > > Dear reviewer jrqf, we sincerely thank you for taking time to read our rebuttal and raise the follow-up questions. We respond to them point by point below:
> > >
> > > ---
> > >
> > > **Question 1**: Gaussian noise/random sampling/random masking is difficult to simulate real point cloud noise
> > >
> > > **Response 1**: We thank the reviewer for this thoughtful observation. We fully agree that real-world point cloud noise can be far more complex than what is captured by simple augmentations such as Gaussian noise, random point dropout, or sampling. Our use of these techniques follows standard practices in existing sim-to-real literature [A, B, C], aiming to improve the robustness of the model and reduce overfitting to idealized simulation data. To validate the real-world performance of our model, we directly deployed the simulation-trained model in real-world dexterous grasping experiments without any fine-tuning or additional refinement. Notably, we did not carefully optimize the camera-object distance to minimize depth noise, nor did we employ any point cloud completion networks. Instead, we used a pre-trained VGGT model to process real-world multi-view RGB images, extracted the target object point clouds and fed them into our grasp transfer model. Our method achieved an average grasp success rate of 70% across 4 object categories and 5 task specifications, demonstrating promising robustness to real-world grasping scenarios. Meanwhile, we appreciate the reviewer’s concern regarding the sim-to-real gap, and we think that exploring more advanced and realistic point cloud simulation techniques [D] is a promising direction to further mitigate this gap and scale up to more complex manipulation scenarios. We will elaborate on this discussion in the revised manuscript. Additionally, we will release our source code, trained models, and real-world evaluation results upon acceptance.
> > >
> > > [A] Torne Marcel, et al. Reconciling Reality through Simulation: A Real-to-Sim-to-Real Approach for Robust Manipulation. RSS 2024.
> > >
> > > [B] Tao Chen, et al. Visual dexterity: In-hand reorientation of novel and complex object shapes. Science Robotics 2023.
> > >
> > > [C] Ruikai Cui, et al. Points2Surf Learning Implicit Surfaces from Point Clouds. ICCV 2023.
> > >
> > > [D] Qiyu Dai, et al. Domain Randomization-Enhanced Depth Simulation and Restoration for Perceiving and Grasping Specular and Transparent Objects. ECCV, 2022.

---

> > > ### Author Response · Authors · 2025-08-05
> > > **Response to new question 2**
> > >
> > > **Question 2**: How should templates be chosen for categories that have never been seen before? Which category do the template hands for these categories (Bowl, Cup, Flashlight, Game controller, Headphones, Knife, Trigger sprayer, and Wine glass) come from, and are all the templates derived from a single object example?
> > >
> > > **Response 2**: We sincerely apologize that our previous rebuttal may not have fully resolved your question. We believe the confusion may stem from a misunderstanding of our experimental setting regarding seen and unseen categories. Specifically, seen categories refer to object categories on which our diffusion-based grasp transfer framework is trained, while unseen categories are those that our model has never seen during training. We evaluate generalization by directly applying the trained model to these novel categories. To clarify, our method assumes access to a single template object from the same category as any target object at test time, no matter whether the category is seen or unseen. For instance, in the case of unseen categories such as Bowl, Cup, Flashlight, Game controller, Headphones, Knife, Trigger sprayer, and Wine glass, we randomly select one object instance from the same category to serve as the template. All template hands are derived from a single object example per category. This setting reflects a practical and realistic scenario where a canonical model or demonstration exists for a functional object category, and grasp knowledge needs to be transferred to novel instances within that category[E, F, G]. Under this setting, our main contribution lies in proposing a task-conditioned diffusion model that explicitly models the geometric relationship between the template and the target object. This enables effective and generalizable dexterous grasp transfer without requiring training on every possible object category. Through extensive experiments across both *seen* and *unseen* object categories, we demonstrate that our framework outperforms existing analytical, generative, and transfer-based approaches in terms of grasp quality and generalization capability. Moreover, we believe that our proposed task-conditioned geometric feature modeling framework also has huge potential for cross-category grasp transfer in the future. For example, by modeling local shape primitives and part-level similarity, our method could enable transferring grasps between objects that share similar functional components, even if they belong to different categories (e.g., transferring from a mug to a teapot). We consider this a promising future direction, and we plan to explore it further using the recent Open-Set Dexterous Grasp Dataset [H]. We will clarify this experimental assumption, discuss its implications, and elaborate on the potential for broader generalization in our revised manuscript. Thank you again for pointing out this important aspect.
> > >
> > > [E] Hongtao Wen, et al. TransGrasp: Grasp Pose Estimation of a Category of Objects by Transferring Grasps from Only One Labeled Instance. ECCV 2022.
> > >
> > > [F] Bowen Wen, et al. You Only Demonstrate Once: Category-Level Manipulation from Single Visual Demonstration. RSS 2022.
> > >
> > > [G] Junzhe Zhu, et al. DenseMatcher: Learning 3D Semantic Correspondence for Category-Level Manipulation from a Single Demo. ICLR 2025.
> > >
> > > [H] Yilin, Wei, et al. Afforddexgrasp: Open-set language-guided dexterous grasp with generalizable-instructive affordance. Arxiv 2025.

---

> > > > ### Comment · Reviewer_jrqf · 2025-08-06
> > > >
> > > > Thank you for the author's reply. If you have used a sample from the same category—even though it was not used in your model training but was still employed during testing—strictly speaking, I do not consider this a completely unseen category. It should instead be classified under the scope of "one-shot". The author should revise the paper to enable readers to understand it more clearly.
> > > >
> > > > Secondly, “All template hands are derived from a single object example per category”. If the same instance is chosen for objects of the same category, I believe this would be overly restrictive in terms of usage and would inevitably lead to bias issues.

---

> > > > > ### Author Response · Authors · 2025-08-07
> > > > >
> > > > > Dear reviewer jrqf, we thank you for your time and the follow-up feedback. For the first point, we would like to clarify that we followed a widely used practice in relevant robotic literature (e.g., [A, B, C]) to divide the testing objects into "seen" and "unseen" objects / categories. It is based on the consideration to comprehensively evaluate the model performance when being directly applied to the novel objects / categories without retraining. We appreciate you pointing out that this terminology could be confusing to readers. It is a terminological issue that can be easily addressed. We will follow your valuable suggestion and revise the manuscript accordingly to ensure clarity. Thanks for the advice.
> > > > >
> > > > > For the second point, we would like to clarify that our model is not confined to a specific template. In our experiment, the template was randomly selected and then fixed. It is based on the consideration to evaluate grasp transfer performance for objects with different shape variations relative to the template and to ensure the reproducibility of all experimental results. We would also like to clarify that apart from the experiments with this fixed template, we have evaluated our model with varying template shapes and template grasps sampled from different methods (as presented in the response to new question 3). These results demonstrate that our model is robust to the choice of template. Therefore, we believe our model will not be overly restrictive in terms of usage and will not suffer from significant bias issues. We sincerely thank you for your response to this point. Our model, the used template and all results from the evaluation of the template choice will be released in the future.
> > > > >
> > > > > [A] One-Shot Learning for Task-Oriented Grasping. RA-L 2023.
> > > > >
> > > > > [B] Robotic Pick-and-Place of Novel Objects in Clutter with Multi-Affordance Grasping and Cross-Domain Image Matching. Int. J. Robot. Res 2019.
> > > > >
> > > > > [C] TD-TOG Dataset: Benchmarking Zero-Shot and One-Shot Task-Oriented Grasping for Object Generalization. Arxiv 2025.

---

> > > ### Author Response · Authors · 2025-08-05
> > > **Response to new question 3**
> > >
> > > **Question 3**: During inference, how should the template hand and object be selected? Is it predefined?
> > >
> > > **Response 3**: We sincerely apologize that our previous rebuttal did not fully resolve your concern. We now provide a more detailed clarification. In our proposed category-level dexterous grasp transfer framework, for each test object, our model requires a template from the same object category as input. The template shape can be retrieved from public 3D assets [I, J] based on the category name, or synthesized using text-to-3D generation methods [K]. For the template hand, we use the dexterous grasp provided by the CapGrasp dataset in our main experiments. However, our framework allows for much more flexible specification of the template hand. It can be manually defined by the user, retargeted from human demonstrations, sampled by analytical methods, or generated by existing generative models.
> > >
> > > We understand the reviewer’s concern about the need for a predefined object template. We would like to clarify that our model only requires one template per category, and this single template is sufficient to handle a wide range of novel objects within that category. We believe this is a practical assumption, especially given the complexity and high DoF of dexterous hands. Importantly, our model is robust to variations in the template input, as demonstrated in the following:
> > >
> > > 1. Template Shape Robustness: We conducted additional experiments on the CapGrasp dataset where the template shape for each category was randomly selected (different from the default template). The average success rate on 8 unseen categories was 75.36% vs. 74.14% (default template), and the consistency score was 77.19% vs. 79.28%, showing a high robustness to the template shape variation.
> > > 2. Task Specification Robustness: Our model generalizes to novel task specifications unseen during training. In the CapGrasp experiment, none of the task descriptions in the test set appear in the training set, yet our model performs robustly on both seen and unseen categories (see Table 2 in the main paper).
> > > 3. Template Hand Source Robustness: In addition to using CapGrasp-provided template hands, we also experimented with template hands generated by an analytical method and a generative model. These methods are task-agnostic, and we evaluated the task-agnostic grasping performance when using them as the template hand. Our model successfully transferred these grasps to novel objects, demonstrating robustness to how the template hand is specified.
> > >
> > > Given these results, we believe our framework offers a practical and generalizable solution for category-level dexterous grasp transfer, with minimal reliance on specific template selection. We will add this clarification and the supporting results to the revised manuscript.
> > >
> > > [I] Chang A. X. et al. ShapeNet: an information-rich 3D model repository. Arxiv 2015.
> > >
> > > [J] Deitke, Matt, et al. Objaverse: A universe of annotated 3d objects. CVPR 2023.
> > >
> > > [K] Meshy: Create stunning 3d models with ai. 2024.

---

> > > ### Author Response · Authors · 2025-08-05
> > > **Response to new question 4**
> > >
> > > **Question 4**: Questions about the additional experiment results for DexGraspNet and ContactGen.
> > >
> > > **Response 4**: We sincerely thank the reviewer for carefully checking our additional results and apologize for not clearly explaining the experimental setup in our previous rebuttal. In the experiments presented in A4, our goal was to test the robustness of our grasp transfer model to the choice of the template hand. To do so, we experimented with two additional and representative approaches to generate template hands on the template shape: (1) DexGraspNet, an analytical, semantic-free method based on force closure, and (2) ContactGen, a generative model trained on seen categories. Then, we directly applied our model without any retraining, transferring the template hands generated by these two methods to various novel objects for dexterous grasping. The evaluation was conducted in a **task-agnostic grasping setting on 16 seen object categories**. Therefore, the result of ‘our original’ matches to the results that we reported in Table 1 of the main paper. This task agnostic setting is compatible with the semantic-free method DexGraspNet. Meanwhile, since the evaluation is on seen categories, we are able to effectively utilize ContactGen to sample template hand around templates belonging to seen categories. The performance of our model when using these alternative template hands demonstrates its robustness to different choices of template hand, including those generated by analytical or generative methods. As such, the comparable performance of ContactGen-based templates does not contradict findings in [2], as they are evaluated here only on seen categories. We will clarify this experimental setup and its intention more explicitly in the revised manuscript.

---

### Official Review · Reviewer_EFeh · 2025-07-03

**Clarity:** 3
**Significance:** 3
**Originality:** 3
**Rating:** 5
**Confidence:** 4

**Summary:**

This paper proposes a conditional diffusion model for dexterous grasp transfer. In detail, the proposed method generates an object contact map and transfers high-quality grasps from shape templates to new objects within the same category using a diffusion-based model, which can enhance the generalization to novel objects. The framework includes an innovative dual mapping mechanism and employs cascaded diffusion models to generate consistent contact, part, and direction maps. A robust optimization scheme recovers grasp configurations, ensuring stability and feasibility. The method achieves good performance in grasp success rates, penetration depths, and object coverage compared with other methods.

**Questions:**

1. Could the authors clarify the sensitivity of the model to variations in template quality or inaccuracies in template selection? (If I have not missed) How were the templates selected in the experiments?

2. How would the method perform under real-world sensor noise and different types of robotic hands?

**Ethical Concerns:**

["NO or VERY MINOR ethics concerns only"]

**Final Justification:**

This paper proposes a new conditional diffusion model for dexterous grasp transfer. The contribution is solid with novel technical components and competitive performance. With the planned revisions and additions, I believe it will be a valuable addition to the community.

**Limitations:**

Yes.

**Paper Formatting Concerns:**

No major formatting concerns.

**Quality:**

3

**Strengths And Weaknesses:**

Strengths:

1. Transferring contact maps through conditional diffusion models is an interesting design and shows good performance.

2. The proposed conditional diffusion framework that jointly generates contact, part, and direction maps can significantly enhance grasp stability and consistency.

3. Experiments show superior performance across multiple metrics compared to existing analytical, generative, and transfer-based approaches.

4. The paper is well-written and easy to follow.

Weaknesses:

1. The performance of the proposed method may depend on the similarity between the template shape and the target object. A further study could be conducted to analyze how varying the templates influences performance. Using templates can provide additional information, simplifying the task compared to methods that do not utilize templates. Experiments could also be carried out either without templates or by introducing templates into generative models for comparative analysis.

2. The proposed method involves cascaded diffusion models and optimization processes, which might pose challenges for downstream tasks, especially robotic grasping tasks with limited computational resources and real-time requirements.

3. The significance of utilizing contact maps in hand-object interaction has been highlighted in prior research. It may be beneficial to include a more detailed discussion and additional references to closely related works. For example, [Ref 1] predicts an object contact map alongside the hand point cloud to model grasping motions and further refines the results through an optimization stage.

[Ref 1] SAGA: Stochastic Whole-Body Grasping With Contact. ECCV 2022.

---

> ### Author Rebuttal · Authors · 2025-07-31
>
> Dear reviewer EFeh, we appreciate the time and effort you dedicated to reviewing our work. Our detailed, point-by-point responses are provided below:
>
> ---
>
> **Q1: How is the shape template selected in the experiments, and how does varying the templates influence the performance?**
>
> **A1**: We thank the reviewer for the thoughtful question and apologize for any confusion. In our experiments, for each object category, one shape template is randomly selected from the CapGrasp dataset and used consistently across all experiments. The same template is also applied to other transfer-based baselines such as Tink to ensure a fair comparison. During training, the template-target object pairs are randomly sampled to encourage diversity, and we further employ standard point cloud augmentation techniques such as random point masking out and additive Gaussian noise to improve the model’s robustness to shape and distribution variations. To further evaluate our model’s sensitivity to the choice of shape template, we conducted additional experiments on 8 unseen object categories using varying template shapes. Specifically, for each target object during testing, we randomly selected a different shape template (from the same category but not the same instance) from CapGrasp, and directly applied our trained model for grasp transfer without any re-training. As shown in the Table, our model maintains strong performance across different template-target combinations, demonstrating that the proposed grasp transfer framework is not restricted to a fixed shape template and is robust to shape variation between the template and the target object.
>
> Table A: Robustness to Template Selection Strategy on Unseen Categories
>
> | Method      | SR (%) $\uparrow$ | Pen. $\downarrow$ | Cov. $\uparrow$ | Cont. Err. $\downarrow$ | Consis. $\uparrow$ |
> | ----------- | :-----------: | :---------------: | :-------------: | :---------------------: | :----------------: |
> | Ours        |     74.14     |       1.36        |      30.05      |         0.0363          |       79.28        |
> | Ours-Random |     75.36     |       1.50        |      32.32      |         0.0391          |       77.19        |
>
> ---
>
> **Q2: Additional comparison with generative models + templates.**
>
> **A2**: We sincerely thank the reviewer for the insightful suggestion to strengthen our experimental comparisons. Following your suggestion, we modified the recent generative model DexGYS [32] to incorporate template information. Specifically, we introduced a new template branch in the first-stage module of DexGYS, which takes both the template object point cloud and the corresponding hand point cloud as inputs. We extracted template-relevant features using a point cloud backbone and concatenated them with the original target object and textual task features to condition the grasp generation process. We re-trained this customized DexGYS+Template model from scratch on the same training dataset used for our method.
>
> Table B: Comparison With Baseline With or Without Template on Unseen Categories
>
> | Method          | SR (%) $\uparrow$ | Pen. $\downarrow$ | Cov. $\uparrow$ | Cont. Err. $\downarrow$ | Consis. $\uparrow$ |
> | --------------- | :-----------: | :---------------: | :-------------: | :---------------------: | :----------------: |
> | DexGYS          |     39.16     |       23.63       |    **32.37**    |         0.1332          |       68.08        |
> | DexGYS+Template |     56.88     |       14.20       |      29.90      |         0.0712          |       70.98        |
> | Ours            |   **74.14**   |     **1.36**      |      30.05      |       **0.0363**        |     **79.28**      |
>
> As shown in the table, incorporating the template grasp improves the performance of DexGYS to some extent, especially in terms of grasp quality and task relevance. However, the performance of this variant remains consistently inferior to our proposed model across all 8 unseen object categories. This result suggests that simply using the template grasp as an additional condition, without explicitly modeling the geometric similarity between the template and the target object, is insufficient for generalizing to novel categories. Furthermore, we qualitatively observe that the improvements from template conditioning in DexGYS are not always consistent, particularly when the template and target objects differ significantly in shape. These results demonstrate the effectiveness of our proposed transfer framework, which explicitly learns task-conditioned template-object geometric similarity and enables robust grasp transfer across diverse objects. Thanks again for this insightful comment to further enhance our experiments. We will include this new experimental comparison and the corresponding quantitative and qualitative results in the revised manuscript.
>
> ---
>
> **Q3: More detailed analysis on real-world noise and different hands.**
>
> **A3**: We thank the reviewer for this important question. In our main paper, we have already demonstrated that our model can be applied to different five-fingered dexterous hands (e.g., the ShadowHand and a human hand model Mano), which showcases the generalization capability of our method across various hand morphologies. To further evaluate the effectiveness of our method under real-world sensor noise and more types of robotic hands, we conducted real-world dexterous grasping experiments on a self-developed humanoid robotic platform equipped with an Inspire dexterous hand mounted at the end of the robot arm. Our visual perception setup used a head-mounted ZED camera and two RealSense cameras on the left and right sides of the robot to capture multi-view RGB images. Then, we leveraged Grounding-DINO to segment the target object from the captured views according to language instruction, and employed the recent 3D foundation model VGGT to reconstruct the target’s point cloud from these segmented views. Without any re-training or fine-tuning of our diffusion model, we directly used the reconstructed point cloud to predict the dexterous grasp configuration. To execute the grasp, we adopted the protocol from [32]. First, the robot arm positioned the hand's root at the predicted 6-DOF pose. The Inspire hand then actuated its joints according to the predicted grasp poses. Due to limited time, our evaluation covered four novel object categories (Bowl, Wineglass, Cup, and Headphones) across five different task specifications. We attempted each grasp five times per object to record the average success rate. A grasp was considered successful if the object was lifted over 30 cm from the table and held securely for five seconds. Our method achieved an overall success rate of 70% across these categories (Bowl: 60%, Wineglass: 80%, Cup: 80%, Headphones: 60%), which highlights its robustness to real-world sensor noise and generalizability to different types of robotic hands. As mentioned in our future work, exploring the framework’s generalization to dexterous hands with varying numbers of fingers and morphological structures is an interesting and open research direction we plan to pursue. Additionally, we will incorporate a thorough description of our robotic platform and the results from the real-world experiments into the revised manuscript. We thank you again for your constructive suggestions.
>
> ---
>
> **Q4: Possible limitation for practical application in robot platforms with limited computational resources.**
>
> **A4**: We thank the reviewer for this very important and practical comment. We agree that deploying computationally intensive models on resource-constrained robots is a critical challenge for real-world application, and we appreciate the constructive, application-focused feedback. Our framework, like many state-of-the-art generative models, is not yet optimized for real-time performance on edge devices, and we see this as an important direction for future engineering work. To bridge this gap for a practical deployment, we would investigate established pathways such as model quantization to reduce the memory footprint and a hybrid cloud/edge computing architecture, where the heavy grasp generation is offloaded to a remote server, leaving the local robot to handle low-latency execution. We will add a discussion of these limitations and potential solutions to the final version of our manuscript. Thank you again for your insightful suggestions.
>
> ---
>
> **Q5: More discussion on works with hand-object interaction.**
>
> **A5**: We thank the reviewer for this good suggestion. We were indeed inspired by prior research such as SAGA and agree that object-centric representations are a highly effective way to model hand-object interaction. Building upon these works, our key contribution is a conditional diffusion model designed to transfer a known, high-quality contact map from a template to a novel object, a task specifically aimed at enhancing generalization. We will add a more detailed discussion contextualizing our work within this line of research to the revised manuscript.

---

> > ### Comment · Reviewer_EFeh · 2025-08-05
> >
> > Thank the authors for the rebuttal. The additional clarifications and experiments effectively address my concerns.

---

> > > ### Author Response · Authors · 2025-08-05
> > > **Thank You!**
> > >
> > > Dear Reviewer EFeh,
> > >
> > > Thank you for your feedback on our rebuttal. We are pleased that our response has addressed your questions. Much appreciate your kind support for our work. Thanks a lot!
> > >
> > > Sincerely,
> > >
> > > Authors

---

### Official Review · Reviewer_UCkK · 2025-07-03

**Clarity:** 3
**Significance:** 3
**Originality:** 2
**Rating:** 5
**Confidence:** 4

**Summary:**

The manuscript presents a transfer-based framework that leverages a conditional diffusion model to migrate high-quality grasps from shape templates to novel objects, conditioned jointly on geometric similarity and free-form task language. Grasp generation is reformulated as the coupled synthesis of an object-centric contact map, part map, and direction map via a dual-branch diffusion network and a cascaded pipeline, which enforces cross-map consistency and yields richer contact semantics. A subsequent optimization stage filters unreliable predictions and reconstructs full hand configurations, producing stable, physically plausible grasps. Experiments on the CapGrasp benchmark indicate that the proposed method achieves superior success rate and contact coverage while reducing penetration relative to analytical, generative, and prior transfer baselines, thereby demonstrating strong generalization to unseen objects, tasks, and categories.

**Questions:**

1. Could the authors incorporate quantitative comparisons with additional grasp-transfer baselines such as DexFuncGrasp and SSCF to contextualize performance gains?
2. Have the authors considered, or can they report, preliminary real-world experiments to substantiate sim-to-real transfer? Even a small-scale study would strengthen the empirical claims.

**Ethical Concerns:**

["NO or VERY MINOR ethics concerns only"]

**Final Justification:**

After reviewing the other reviewers’ comments and the authors’ rebuttal, I have decided to raise my score to Accept. I sincerely thank the authors for their patient, detailed, and rigorous responses.

My two main concerns—(1) comparisons with additional baselines and (2) real-world experiments—have been fully addressed in the rebuttal. I look forward to seeing the comparison details and hardware experiment results incorporated into the revised manuscript, as well as the authors’ future work on the humanoid robotic platform.

**Limitations:**

yes

**Paper Formatting Concerns:**

None.

**Quality:**

3

**Strengths And Weaknesses:**

## Strengths

1. **Well-motivated problem and principled solution.** The paper tackles the salient challenge of transferring dexterous grasps to novel objects and proposes a coherent, diffusion-based framework that is both conceptually sound and technically appealing.
2. **Comprehensive evaluation.** Ablation studies isolate the contributions of each module, and benchmark comparisons substantiate the effectiveness of the overall system.
3. **Clarity of presentation.** The methodology, optimization pipeline, and experimental protocol are described with sufficient detail, making the work straightforward to follow.

## Weaknesses

1. **Limited comparison with closely related work.** Several pertinent approaches—e.g., [66] for pose transfer and [i] for implicit contact-region representation—are not included in the quantitative or qualitative analysis.
2. **Lack of hardware validation.** All results are reported in simulation; the absence of real-robot experiments leaves the practical robustness of the method unverified.
3. **Static outcome.** The framework outputs single grasp configurations only and does not address the synthesis of full action sequences that might be required for dynamic manipulation.

[i] Huang, Zeyu, et al. "Spatial and surface correspondence field for interaction transfer." ACM Trans. Graph, 2024

---

> ### Author Rebuttal · Authors · 2025-07-31
>
> Dear reviewer UCkK, we are grateful for your time and supportive comments highlighting our method as a "*principled solution*" that is both "*conceptually sound*" and "*technically appealing*" to a "*well-motivated problem*". We are also glad you recognized our "*comprehensive evaluation*" and the "*clarity of presentation*". We have provided detailed responses to each of your questions below:
>
> ---
>
> **Q1: Comparison with the additional pose transfer approach DexFuncGrasp, and contact-region representation approach SSCF.**
>
> **A1**: We sincerely thank the reviewer for pointing out these two relevant works. Following your suggestion, we have expanded our experiments to include comparisons with both DexFuncGrasp and SSCF. For DexFuncGrasp, which transfers grasps via shape interpolation between the template and target object, we note that its transfer strategy is conceptually similar to the transfer-based method Tink that we have evaluated in our main paper. DexFuncGrasp requires category-specific training to ensure interpolation quality, and thus cannot generalize to unseen object categories. To ensure fairness, we compared our method with DexFuncGrasp on various novel objects from the 16 seen categories and evaluated both human and robotic hand grasp transfers. As reported in Table A and B, our method consistently outperforms DexFuncGrasp in both settings. Furthermore, unlike DexFuncGrasp, our method does not require retraining and can directly transfer grasps to unseen object categories.
>
> Table A: Human Grasp Transfer Comparison on Seen Categories
>
> | Method       | Pen. Volume $\downarrow$ | Contact Ratio $\uparrow$ | Simulation Disp. $\downarrow$ |
> | ------------ | :----------------------: | :----------------------: | :---------------------------: |
> | Dexfuncgrasp |           1.91           |          77.62%          |             3.30              |
> | Ours         |           1.11           |          87.45%          |             2.76              |
>
> Table B: Robotic Grasp Transfer Comparison on Seen Categories
>
> | Method       | Pen. $\downarrow$ | Contact Ratio $\uparrow$ | SR (%) $\uparrow$ |
> | ------------ | :---------------: | :----------------------: | :---------------: |
> | Dexfuncgrasp |       1.14        |          25.13%          |       69.82       |
> | Ours         |       1.47        |          38.16%          |       84.65       |
>
> For SSCF, which models contact regions via a learned spatial and surface correspondence field, we found that it requires a separately trained model for each object category. Moreover, its official GitHub repository does not provide training code, and only a pre-trained model for the ‘mug’ category is available. Therefore, we conducted a focused comparison on the ‘mug’ category. As shown in Table C, our method can achieve superior performance when compared with SSCF, even though their model was specifically trained for that category. These results highlight the superior generalization capability and effectiveness of our proposed approach compared to both DexFuncGrasp and SSCF. We thank the reviewer again for the valuable suggestions and will include these additional experimental results and analyses in the revised manuscript.
>
> Table C: Human Grasp Transfer Comparison on the 'Mug' Category
>
> | Method | Pen. Volume $\downarrow$ | Contact Ratio $\uparrow$ | Simulation Disp. $\downarrow$ |
> | ------ | :----------------------: | :----------------------: | :---------------------------: |
> | SSCF   |           0.53           |          54.24%          |             4.41              |
> | Ours   |           0.46           |          56.70%          |             3.63              |
>
> ---
>
> **Q2: Real-world robot experiments.**
>
> **A2**: We thank the reviewer for the constructive suggestion. Following your advice, we conducted real-world dexterous grasping experiments to evaluate the effectiveness of our method in practical scenarios. The experiments were performed on a self-developed humanoid robotic platform equipped with an Inspire dexterous hand mounted at the end of the robot arm. For visual perception, we deployed one ZED camera on the robot’s head and placed two RealSense cameras on the left and right sides of the robot to capture multi-view RGB images. Given a textual task instruction, we used Grounding-DINO to segment the target object and leveraged the recent 3D foundation model VGGT to reconstruct the object’s point cloud from the captured views. Without any re-training or fine-tuning of our model, we directly used the reconstructed point cloud to predict the dexterous grasp configuration. For grasp execution, we followed the protocol in [32], where the robot arm first moves to the predicted 6-DOF pose of the hand’s root, and then the Inspire hand actuates its joint angles based on the predicted contact poses. Due to limited time, we tested four novel object categories (i.e., Bowl, Wineglass, Cup, and Headphones) across five different task specifications. For each object, we repeated the grasping process five times and recorded the average grasp success rate. A successful grasp is defined as lifting the object at least 30 cm from the table and holding it stably for 5 seconds. Overall, our method achieved an average success rate of 70% across the tested categories (60% for Bowl, 80% for Wineglass, 80% for Cup, and 60% for Headphones), demonstrating the practical applicability of our approach and its robustness to real-world observation noise. We also observed two typical failure patterns in our real-world experiments: (1) collisions during the arm's approach to the object, often due to motion planning limitations, and (2) unintended contact with the table or object during finger closing, especially for grasping the rim of the cup or bowl. These issues highlight the gap between dexterous grasp generation and its robust execution in practice, which we identify as a key area for future exploration. Also, we will include a detailed description of the robot hardware setup and real-world experimental results in the revised manuscript. Thanks again for your valuable feedback.
>
> ---
>
> **Q3: Only output a single grasp configuration while the dynamic manipulation may require full action sequences.**
>
> **A3**: We greatly appreciate the reviewer’s insightful comment, which well aligns with our observations during the real-world grasping experiments. Predicting a high-quality object-centric dexterous grasp pose is only the first step toward generalizable and stable dexterous manipulation. In real deployments, due to the high degrees of freedom and precision requirements of dexterous robotic hands, executing even a high-quality grasp can be challenging and often prone to failure. Generating full action sequences for dexterous grasp execution remains an open and highly challenging problem, receiving increasing attention in both reinforcement learning (RL) and vision-language-action (VLA) modeling communities. In future work, we are interested in exploring how the generalizable, task-conditioned object-centric representation produced by our transfer framework can be used to guide the training of RL policies or VLA models for sequential dexterous manipulation. Moreover, while our current model outputs a single grasp pose, apart from its use for object grasping, it provides a strong prior about the intended robot-object interaction and the preferred spatial hand-object relationship. This information can be valuable for many other robot-relevant scenarios, such as guiding pre-grasp planning, reachability analysis, safety-aware interaction design, and downstream policy learning. Thanks again for this insightful comment. We will further elaborate on this discussion in our revised manuscript.

---

> > ### Comment · Reviewer_UCkK · 2025-08-08
> >
> > After reviewing the other reviewers’ comments and the authors’ rebuttal, I have decided to raise my score to Accept. I sincerely thank the authors for their patient, detailed, and rigorous responses.
> >
> > My two main concerns—(1) comparisons with additional baselines and (2) real-world experiments—have been fully addressed in the rebuttal. I look forward to seeing the comparison details and hardware experiment results incorporated into the revised manuscript, as well as the authors’ future work on the humanoid robotic platform.

---

> > > ### Author Response · Authors · 2025-08-08
> > > **Thank You!**
> > >
> > > Dear Reviewer UCkK,
> > >
> > > Thank you for your positive feedback on our rebuttal. We are delighted our response addressed your questions. We will follow your suggestion to incorporate the additional comparisons and the results from our real-world experiments into the final version. We sincerely appreciate your kind support for our work and your decision to raise the score!
> > >
> > > Sincerely,
> > >
> > > Authors

---

### Official Review · Reviewer_2Xnf · 2025-07-03

**Clarity:** 3
**Significance:** 2
**Originality:** 3
**Rating:** 4
**Confidence:** 4

**Summary:**

The paper tackles the long-standing generalization gap in dexterous grasp synthesis.
Instead of learning a grasp distribution from scratch for every object, the authors propose to transfer analytically generated, high-quality grasps from a shape template to novel objects of the same category with a cascaded conditional diffusion pipeline.
Key ingredients are:
•	a dual-branch diffusion model that simultaneously reconstructs the template contact map and denoises the target map, connected through bidirectional adaptation modules that encode template–target similarity ;
•	extension from contact map to part and direction maps and a cascaded generation scheme that keeps these three modalities self-consistent ;
•	a robust grasp recovery routine that filters noisy map predictions and optimizes a grasp with differentiable kinematics and force-closure regularizers .
On the large CapGrasp benchmark (16 seen, 8 unseen categories), the method reaches analytical-level success rate (84.7 %) while outperforming recent diffusion and transfer baselines in penetration and coverage (Table 1) and retains clear advantages on unseen categories and language-conditioned tasks (Table 2) . Thorough ablations quantify the contribution of each module .

**Questions:**

1.	Inference latency. How long does a single grasp prediction (contact + part + direction + optimization) take on GPU/CPU? A table comparing to analytical and generative baselines would clarify practical impact.
2.	Template acquisition. In a real deployment, how would you choose or construct the shape template for a new object?
3.	Robustness to partial / noisy observations. Have you tested the transfer when the target point cloud is sparse or occluded? If not, please discuss expected behavior.
4.	Failure analysis. Please include qualitative examples where the transfer fails (large geometric discrepancy, unusual tasks) and discuss why.

**Ethical Concerns:**

["NO or VERY MINOR ethics concerns only"]

**Final Justification:**

The rebuttal addressed several of my key concerns:

Template acquisition: The authors provided a clear and practical strategy for maintaining and expanding a category-level shape template library, including the use of large-scale generative 3D models for unseen categories. This alleviates my earlier concern about template availability in realistic deployments.

Robustness: Additional experiments with partial/noisy point clouds—both in simulation and with real-world reconstructions—show the method retains strong performance, with only a modest drop in success rate. This partially resolves my concern about sensitivity to observation quality.

Inference speed: Detailed timing results and comparisons against analytical, generative, and transfer baselines clarify that the method achieves a favorable balance between quality and runtime, and is competitive for practical use.

Failure analysis: While visual examples are still missing (due to rebuttal-phase constraints), the authors gave a clear conceptual discussion of common failure modes, which they plan to include in the final version.

Minor issues: Typos and formatting will be fixed.

Remaining limitations:

Physical robot experiments are still absent; although real-world point cloud tests help, full hardware validation would strengthen claims.

Some dependency on category-level templates remains, and while the library-based approach is practical, mismatches in topology/semantics still cause failures.

Weighting: The method is technically sound, original in framing grasp transfer as cascaded multi-map diffusion, and empirically strong in simulation with additional robustness validation. While real robot validation is missing, the rebuttal resolves most of my practical concerns and clarifies feasibility.

**Limitations:**

The authors acknowledge evaluation on a single five-finger hand and purely simulated setup; extending to varied embodiments and hardware experiments remains future work . I agree and would add the reliance on template availability as another practical limitation.

**Paper Formatting Concerns:**

1.	Spelling typo in section header “Methdology”

**Quality:**

3

**Strengths And Weaknesses:**

## Strengths
The originality is good. Framing grasping as contact-map transfer with a dual-branch diffusion and a cascaded multi-map design is novel relative to prior diffusion graspers that work in pose space or single-map space. The empirical result is good. Solid architecture description, clear losses, and extensive ablations (e.g., removing adaptation drops SR from 79 → 38 %).
## Weaknesses

1.Simulation-only validation. All quantitative results are in IsaacGym; no physical robot experiments or real point-cloud noise study.

2.Computation cost. Training takes 24 h on two 3090s with 1 000 diffusion steps and batch 56, and inference speed is not reported.

3.Template dependence. The pipeline assumes access to a suitable shape template; the paper does not discuss template selection or failure cases when templates are sparse or mismatched.

---

> ### Author Rebuttal · Authors · 2025-07-31
>
> Dear reviewer 2Xnf, thanks for your time to review our submission and provide constructive comments. In the following, we respond to your questions in detail one by one:
>
> ---
>
> **Q1: Clarify how to acquire the shape template used for grasp transfer.**
>
> **A1**: We thank the reviewer for raising this important question. In our practical implementation, we maintain a shape template for each object category by collecting representative 3D mesh models from publicly available object datasets and asset libraries (e.g., [A, B]), thereby forming a library of shape templates across categories. During inference, we retrieve the corresponding shape template from the library based on the textual task description and the involved object category name. For object categories that are not included in the library beforehand, we follow the approach of [C] to leverage a large-scale pre-trained 3D generative model (e.g., [D]) to generate the required shape template based on the category-level text description. Since our proposed grasp transfer framework operates at the category level, it only requires a single canonical shape template to handle a variety of instances with different shapes and sizes within the same category. This makes the cost of maintaining such a template library affordable and scalable. Moreover, thanks to the expressiveness of the diffusion-based generative model and our design of language-conditioned geometric similarity modeling, our method does not rely on texture information and exhibits robustness to shape variations in the template. This enables us to easily expand the template library using 3D generative models and ensures applicability across a wide range of object categories. We will append these implementation details in our revised manuscript.
>
> [A] Chang A. X. et al. ShapeNet: an information-rich 3D model repository. Arxiv 2015.
>
> [B] Deitke, Matt, et al. Objaverse: A universe of annotated 3d objects. CVPR 2023.
>
> [C] Wang, Zhenwei, et al. Phidias: A Generative Model for Creating 3D Content from Text, Image, and 3D Conditions with Reference-Augmented Diffusion. ICLR 2025.
>
> [D] Meshy: Create stunning 3d models with ai. 2024.
>
> ---
>
> **Q2: Robustness to object point cloud artifacts in simulation and real-world settings.**
>
> **A2**: We appreciate the reviewer’s insightful comment. Before reporting our robustness analysis, we would like to clarify that during training, we applied standard point cloud data augmentation techniques, including random point sampling and additive Gaussian noise in the 3D coordinates, to improve the grasp transfer model’s robustness to real-world point cloud artifacts. Following the reviewer’s suggestion, we conducted robustness evaluations in both simulation and real-world settings. In simulation, we followed [A] to randomly mask out 30% of the object points to simulate irregular and incomplete point cloud distributions. Additionally, we added Gaussian noise with a standard deviation of 0.03 to the remaining points to emulate potential sensor-induced coordinate errors. As shown in the table below, our grasp transfer model remains robust under such conditions, with no significant drop in quantitative grasp quality metrics.
>
> Table A: Quantitative Results of Robustness to Simulated Point Cloud Noise
>
> | Test Data | SR (%) $\uparrow$ | Pen $\downarrow$ | Cov $\uparrow$ | Cont. Err. $\downarrow$ | Consis. $\uparrow$ |
> | --------- | :-----------: | :--------------: | :------------: | :---------------------: | :----------------: |
> | Clean     |     74.14     |       1.36       |     30.05      |         0.0363          |       79.28        |
> | Noisy     |     69.34     |       2.67       |     24.19      |         0.0370          |       74.13        |
>
> While the grasp success rate slightly decreased from 74.14% to 69.34%, we found this was primarily due to missing points in critical object regions (e.g., the handle of a mug or the rim of a bowl) that can impair accurate contact computation during the final optimization step. In our real-world experiments, we reconstructed object point clouds from multi-view RGB images captured by one ZED camera and two RealSense cameras. Due to partial occlusion and limited sensor coverage, the reconstructed point clouds exhibited realistic noise and missing regions. We directly fed these point clouds into our grasp transfer pipeline and executed the transferred configuration on a dexterous robotic hand. The results show that our method achieves an average grasp success rate of 70% under real-world point cloud artifacts, demonstrating strong robustness. Due to the page limit, please refer to our response to Q2 of reviewer UCkK for further details on real-world experiments.
>
> [A] Wang, Tao, et al. Sparse convolutional networks for surface reconstruction from noisy point clouds. WACV 2024.
>
> ---
>
> **Q3: More detailed analysis on the inference speed.**
>
> **A3**: We thank the reviewer for the valuable suggestion. To provide a detailed breakdown of the inference speed, we conducted repeated grasp transfer experiments on 10 different objects across 50 task specifications. All experiments were performed on a workstation equipped with an Intel(R) Xeon(R) Gold 6326 CPU and an NVIDIA RTX 3090 GPU (24 GB). On average, our method requires 14.5 seconds for contact map transfer, 15.8 seconds for part map transfer, and 16.3 seconds for direction map transfer. These steps are followed by a final grasp optimization stage, which takes approximately 15.6 seconds. We also compared the overall inference time of our method with two analytical approaches, two generative baselines, and one transfer-based baseline.
>
> Table B: Quantitative Comparison of Inference Time for Grasp Generation
>
> | Method                |  DFC  | DexGraspNet | ContactGen | UGG  | Tink | Ours-Contact | Ours |
> | --------------------- | :---: | :---------: | :--------: | :--: | :--: | :----------: | :--: |
> | Time (s) $\downarrow$ | >1000 |     > 400     |    17.1    | 76.0 | 87.6 |     ~30.1     | 62.2 |
>
> As shown in the table, our method generates high-quality grasps more than 10× faster than analytical methods, while achieving comparable or even superior grasping quality and success rates (i.e., 84.65 for ours vs. 78.98 for DFC and 83.64 for DexGraspNet). Compared to generative methods, our framework operates at a similar inference speed but significantly outperforms them in grasp quality and success rate (i.e., 84.65 for ours vs. 73.00 for ContactGen and 70.50 for UGG). Overall, these results indicate that our method strikes a favorable balance between inference speed and grasping performance. We believe this makes our framework a strong foundation for further optimization and deployment in practical, time-sensitive robotic applications. We will report the detailed inference speed of our method in the revised manuscript.
>
> ---
>
> **Q4: Possible failure analysis and discussion.**
>
> **A4**: We thank the reviewer for this important question. Due to policy constraints this year, we are unfortunately unable to provide qualitative visualizations of failure cases during the rebuttal phase. However, we will elaborate on these cases with qualitative examples and detailed discussion in the revised manuscript. Within our framework, the textual task specification is used not to directly predict the grasp pose, but rather as a task-relevant condition to guide the modeling of geometric similarity between the shape template and the target object. This design enhances the model’s robustness to variations in task instructions, especially compared to conventional generative methods that map text directly to grasp configurations (as demonstrated in Table 3 of the main paper). That said, failure cases do exist in our grasp transfer framework, particularly when object instances within the same category exhibit not only shape variations but also significant differences in topology or structural composition. In such cases, the success of grasp transfer depends not only on geometric similarity but also on whether the task-relevant affordance region lies within or outside the topologically mismatched area. For example, within the 'camera' category, transferring a grasp from a DSLR camera (with a protruding lens) to a compact camera (without a lens) works well for the task “to press the camera's button,” since the relevant affordance region is preserved. However, for the task “to hold the camera by its lens,” the transfer fails due to the absence of a corresponding semantic region on the target object. These limitations highlight a potential weakness in our current approach when handling large topological or semantic mismatches. In future work, we plan to incorporate object-centric semantic primitives to enhance the model’s ability to accommodate such structural variations. Thanks again for this constructive comment, and we will expand the discussion on failure cases in our revised manuscript.
>
> ---
>
> **Q5: Spelling typo in section header.**
>
> **A5**: Thanks for your careful checking. We will proofread the manuscript thoroughly and correct all spelling and grammatical issues in the revised version.

---

> ### Comment · Area_Chair_76Jy · 2025-08-05
>
> Dear Reviewer 2Xnf,
>
> The author-rebuttal phase is now underway, and the authors have provided additional clarifications and performance results in their rebuttal. Could you please take a moment to review their response and engage in the discussion? In particular, we’d appreciate your thoughts on whether their revisions adequately address your initial concerns. Thank you for your time and valuable contributions.
>
> Best,
> Your AC

---

### Note · Authors · 2025-08-14

Dear Reviewers and Area Chair,

We would like to express our sincere gratitude for your valuable suggestions and insights. We appreciate all of you for your supportive comments highlighting our work's strengths:

**1. Clear motivation** (Reviewers UCkK, jrqf)

**2. Novel method & conceptually sound framework** (Reviewers 2Xnf, UCkK, EFeh)

**3. Superior performance & comprehensive evaluation** (Reviewers 2Xnf, UCkK, EFeh)

**4. Clear presentation & well-written paper** (All Reviewers)

Building on this positive feedback, we also thank the reviewers for their constructive suggestions, which we have diligently addressed as follows:

1. **Provided more results on real-world robot experiments**. We conducted experiments on a humanoid robot with a dexterous hand, using a vision-based pipeline for 3D reconstruction. Crucially, without any fine-tuning, our model achieved a 70% average success rate across four novel object categories, demonstrating its strong practical applicability and robustness to real-world observation noise.
2. **Analyzed the model's robustness to template selection.** We showed that our model is highly robust to the choice of template, maintaining strong performance with randomized template shapes, unseen task specifications, and template grasps from different sources.
3. **Addressed other points and clarifications**. Following the reviewers' suggestions, we have incorporated additional comparisons, introduced new generative evaluation metrics, and clarified several other points in the rebuttal.

We are pleased that our rebuttal has ‘fully’ and ‘effectively’ addressed the concerns of Reviewer UCkK and EFeh, and are particularly encouraged by Reviewer UCkK's willingness to raise the score to 'Accept'. We will polish our paper in the revised version for improved clarity:

1. We will integrate all rebuttal experiments and analyses, including the real-world robot results, template robustness tests, failure case studies, and additional comparisons. We will also thoroughly polish the writing to ensure a higher level of clarity and quality.
2. We will release our code, models, and data to ensure full reproducibility for the community.

Finally, we are sincerely grateful that **all reviewers** have recognized the contributions of our work and **provided a positive assessment**. We are confident our contributions will inspire valuable future work and are optimistic about the potential impact on the community.

---

### Decision · Program_Chairs · 2025-09-17

**Decision:**

Accept (poster)

**Comment:**

The paper presents a conditional diffusion model for transferring high-quality dexterous grasps from shape templates to novel objects within the same category, aiming to address generalization challenges in grasp synthesis. The reviewers acknowledged several strengths of the proposed method: 1) a novel contact-map transfer strategy for grasp generation using dual-branch diffusion and cascaded multi-map design; 2) a well-motivated and principled solution with a coherent framework; 3) strong results across multiple metrics compared to existing approaches, with significant empirical improvements and comprehensive ablation studies; and 4) clear presentation. However, they also raised significant concerns initially regarding: 1) limited validation of real-world noise conditions and physical robot experiments, 2) unclear computational complexity in inference, 3) heavy reliance on shape templates without conducting robustness evaluation, and 4) insufficient comparison with related work and lacking generative metrics.

During the author-reviewer discussion, the authors provided a detailed response that included more results on real-world robot experiments, inference speed comparisons, analysis of the model's robustness to template selection, and additional comparisons with other methods using generative metrics. The rebuttal addressed most of the reviewers' concerns, and all four reviewers responded positively to the discussion. In particular, Reviewers UCkK and EFeh felt their major concerns were satisfactorily addressed and recommended acceptance, while the other two reviewers had remaining concerns regarding points 1) and 3) but maintained a positive assessment.

The AC agrees with the reviewers that the authors' rebuttal effectively addressed the main weaknesses of the initial submission and that the strengths of the paper outweigh its weaknesses. Overall, the AC believes the work is a valuable contribution to the field and therefore recommends acceptance. The authors should revise the manuscript to incorporate the reviewers' feedback and address the points discussed in the rebuttal.